# Trajectory-based differential expression analysis for single-cell sequencing data

Koen Van den Berge [1,2,3], Hector Roux de Bézieux[4,5], Kelly Street[6,7], Wouter Saelens [1,8], Robrecht Cannoodt [8,9,10], Yvan Saeys [1,8], Sandrine Dudoit[3,4,5,11✉] & Lieven Clement [1,2,11✉]

Trajectory inference has radically enhanced single-cell RNA-seq research by enabling the study of dynamic changes in gene expression. Downstream of trajectory inference, it is vital to discover genes that are (i) associated with the lineages in the trajectory, or (ii) differentially expressed between lineages, to illuminate the underlying biological processes. Current data analysis procedures, however, either fail to exploit the continuous resolution provided by trajectory inference, or fail to pinpoint the exact types of differential expression. We introduce tradeSeq, a powerful generalized additive model framework based on the negative binomial distribution that allows flexible inference of both within-lineage and between-lineage differential expression. By incorporating observation-level weights, the model additionally allows to account for zero inflation. We evaluate the method on simulated datasets and on real datasets from droplet-based and full-length protocols, and show that it yields biological insights through a clear interpretation of the data.

[1] Department of Applied Mathematics, Computer Science and Statistics, Ghent University, Ghent, Belgium. [2] Bioinformatics Institute Ghent, Ghent University, Ghent, Belgium. [3] Department of Statistics, University of California, Berkeley, CA, USA. [4] Division of Biostatistics, School of Public Health, University of California, Berkeley, CA, USA. [5] Center for Computational Biology, University of California, Berkeley, CA, USA. [6] Department of Data Sciences, Dana-Farber Cancer Institute, Boston, MA, USA. [7] Department of Biostatistics, Harvard T.H. Chan School of Public Health, Boston, MA, USA. [8] Data mining and Modelling for Biomedicine, VIB Center for Inflammation Research, Ghent, Belgium. [9] Center for Medical Genetics, Ghent University Hospital, Ghent, Belgium. [10] Department of Biomolecular Medicine, Ghent University, Ghent, Belgium. [11]These authors contributed equally: Sandrine Dudoit, Lieven Clement. ✉email: sandrine@stat.berkeley.edu; lieven.clement@ugent.be

Single-cell transcriptome sequencing (scRNA-seq) has revolutionized modern biology by allowing researchers to profile transcript abundance at the resolution of an individual cell. This has opened new avenues of research to study cellular pathways during the cell cycle, cell-type differentiation, or cellular activation. Indeed, scRNA-seq can provide a snapshot of the transcriptome of thousands of single cells in a cell population, which are each at distinct points of the dynamic process under study. This wealth of transcriptional information, however, presents many data analysis challenges. Until recently, statistical and computational efforts have focused mostly on trajectory inference (TI) methods, which aim to first allocate cells to lineages and then order them based on pseudotimes within these lineages. A wide range of TI methods have been proposed; 45 of which are extensively benchmarked in Saelens et al.[1]. Note that we use the term 'trajectory' to refer to the collection of 'lineages' for the process under study.

Most TI methods share a common workflow: dimensionality reduction followed by inference of lineages and pseudotimes in the reduced dimensional space[2]. In that reduced dimensional space, a cell's pseudotime for a given lineage is the distance, along the lineage, between the cell and the origin of the lineage. As such, while pseudotime can be interpreted as an increasing function of true chronological time, there is no guarantee that the two follow a linear relationship. Recent developments have allowed the inference of complex trajectories[3–5]. These advances enable researchers to study dynamic biological processes, such as complex differentiation patterns from a progenitor population to multiple differentiated cellular states[6,7], and have the promise to provide transcriptome-wide insights into these processes.

Unfortunately, statistical inference methods are lacking to identify genes associated with lineage differentiation and to unravel how their corresponding transcriptional profiles are driving the dynamic processes under study. Indeed, differential expression (DE) analysis of individual genes along lineages is often performed on discrete groups of cells in the developmental pathway, e.g., by comparing clusters of cells along the trajectory or clusters of differentiated cell types. Such discrete DE approaches do not exploit the continuous expression resolution that can be obtained from the pseudotemporal ordering of cells along lineages provided by TI methods. Moreover, comparing cell clusters within or between lineages can obscure interpretation: it is often unclear which clusters should be compared, how to properly combine the results of several pairwise cluster comparisons, or how to account for the fact that not all of these comparisons are independent of each other.

A number of methods have been developed for the analysis of bulk RNA-seq time-series data, which can exploit the temporal resolution of samples assayed at different times[8–10]. However, in scRNA-seq, the relationship between gene expression and pseudotime is more complex. In addition, the pseudotimes are continuous, and cells are never at the exact same pseudotime value.

A few methods have been published with the aim of improving trajectory-based differential expression analysis by modeling gene expression as a smooth function of pseudotime along lineages. Monocle[11] tests whether gene expression is associated with pseudotime by fitting additive models of gene expression as a function of pseudotime. However, the method can only handle a single lineage. A similar approach has been adopted by TSCAN[12]. GPfates[4] relies on a mixture of overlapping Gaussian processes[13], where each component of the mixture model represents a different lineage. For each gene, the method tests whether a model with a bifurcation significantly increases the likelihood of the data as compared with a model without a bifurcation, essentially testing whether gene expression is differentially associated with the two lineages. Similarly, the BEAM approach in Monocle 2[5]

allows users to test whether differences in gene expression are associated with particular branching events on the trajectory. These trajectory-based methods improve upon discrete cluster-based approaches by (1) exploiting the continuous expression resolution along the trajectory and (2) comparing lineages using a single test based on entire gene expression profiles. However, both GPfates and Monocle 2 lack interpretability, as they cannot pinpoint the regions of the gene expression profiles that are responsible for the differences in expression between lineages. Moreover, the GPfates model is restricted to trajectories consisting of just one bifurcation, essentially precluding its application to biological systems with more than two lineages (i.e., a multifurcation or more than one bifurcation). BEAM is restricted to the few dimensionality reduction methods that are implemented in the Monocle 2 software, namely, independent component analysis (ICA) and DDRTree[5]. Hence, novel methods to infer differences in gene expression patterns within or between transcriptional lineages with complex branching patterns are vital to further advance the field.

In this paper, we introduce tradeSeq, a method and software package for trajectory-based differential expression analysis for sequencing data. tradeSeq provides a flexible framework that can be used downstream of any dimensionality reduction and TI method. Unlike previously proposed approaches, tradeSeq provides several tests that each identify a distinct type of differential expression pattern along a lineage or between lineages, leading to clear interpretation of the results (Fig. 1). In practice, tradeSeq infers smooth functions for the gene expression measures along pseudotime for each lineage using generalized additive models and tests biologically meaningful hypotheses based on parameters of these smoothers. By allowing cell-level weights for each individual count in the gene-by-cell expression matrix, tradeSeq can handle zero inflation, which is essential for dealing with dropouts in full-length scRNA-seq protocols[14]. As it is agnostic to the dimensionality reduction and TI methodology, the approach scales from simple to complex trajectories with multiple bifurcations: tradeSeq only requires the original expression count matrix of the individual cells, estimated pseudotimes, and a hard or soft assignment (weights) of the cells to the lineages to infer the lineage-specific smoothers. We benchmark our method against current state-of-the-art methods using simulated data sets (with cyclic, bifurcating, and multifurcating trajectories) and demonstrate its functionality and versatility on four real data sets. These case studies highlight the enhanced interpretability of tradeSeq's results, which lead to improved understanding of the underlying biology.

## Results

**Statistical model and inference using tradeSeq.** We model gene expression measures as nonlinear functions of pseudotime using a generalized additive model (GAM). In the GAM, each lineage is represented by a separate cubic smoothing spline, and the flexibility of GAMs allows us to adjust for other covariates or confounders as fixed effects in the model. The read counts $Y_{gi}$, for a given gene $g \in \{1, \ldots, G\}$ across cells $i \in \{1, \ldots, n\}$ are modeled using a negative binomial GAM (NB-GAM) with cell and gene-specific means $\mu_{gi}$ and gene-specific dispersion parameters $\phi_g$:

$$\begin{cases} Y_{gi} \sim NB(\mu_{gi}, \phi_g) \\ \log(\mu_{gi}) = \eta_{gi} \\ \eta_{gi} = \sum_{l=1}^{L} s_{gl}(T_{li})Z_{li} + \mathbf{U}_i \boldsymbol{\alpha}_g + \log(N_i). \end{cases} \quad (1)$$

The gene-wise additive predictor $\eta_{gi}$ consists of lineage-specific smoothing splines $s_{gl}$, that are functions of pseudotime $T_{li}$, for lineages $l \in \{1, \ldots, L\}$. The binary matrix $\mathbf{Z} = (Z_{li} \in \{0, 1\}$:

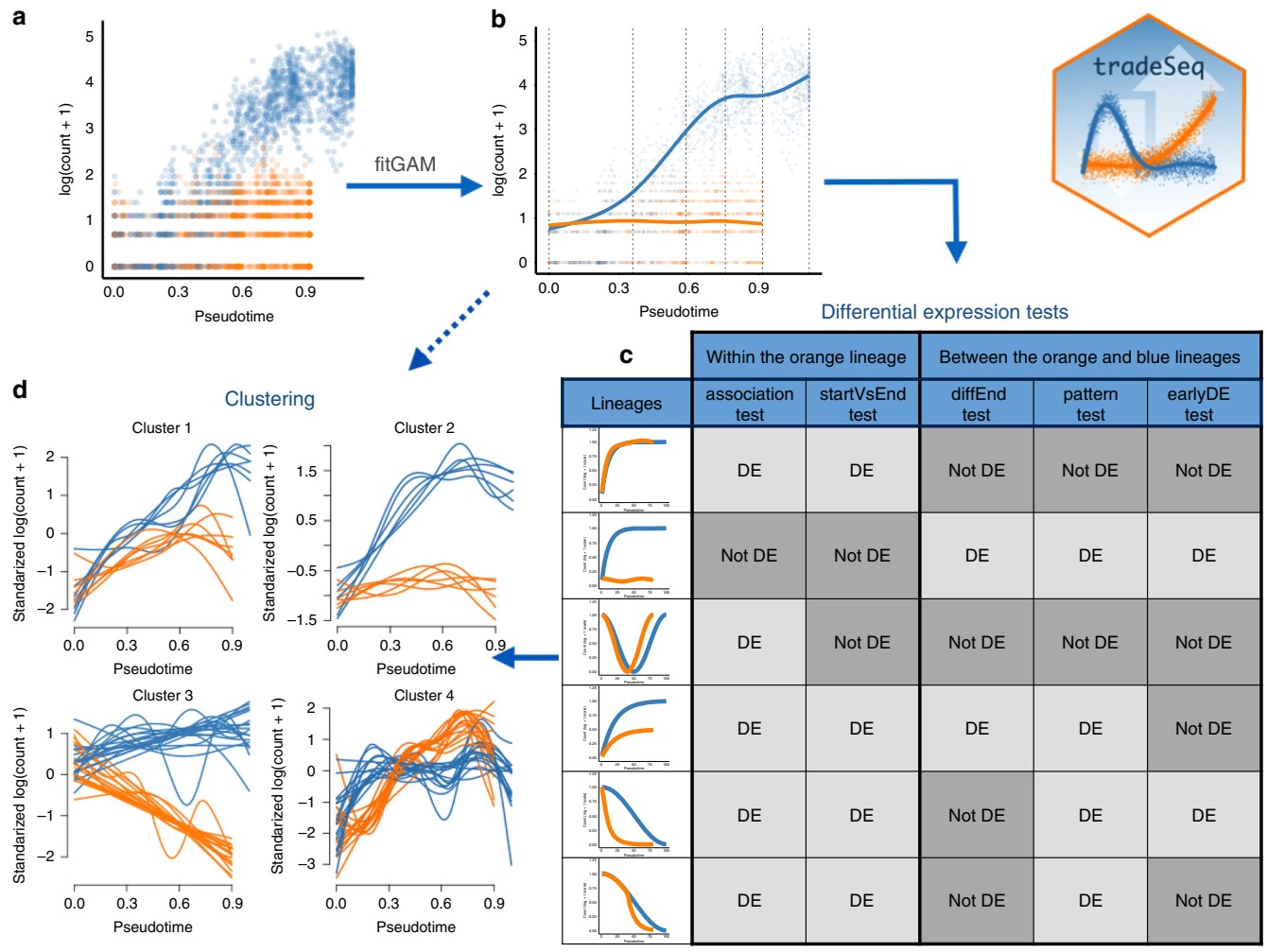

**Fig. 1 Overview of tradeSeq functionality. a** A scatterplot of expression measures vs. pseudotimes for a single gene, where each lineage is represented by a different color (top left). **b** A negative binomial generalized additive model (NB-GAM) is fitted using the `fitGAM` function. The locations of the knots for the splines are displayed with gray dashed vertical lines. **c** The NB-GAM can then be used to perform a variety of tests of differential expression within or between lineages. In the table, we assume that the `earlyDETest` is used to assess differences in expression patterns early on in the lineage, e.g., with option `knots = c(1, 2)`, meaning that we test for differential patterns between the first and second dashed gray lines from panel (**b**). **d** Interesting genes can finally be clustered to display the different patterns of expression.

$l \in \{1, \ldots, L\}$, $i \in \{1, \ldots, n\}$) assigns every cell to a particular lineage based on user-supplied weights (e.g., from slingshot[3] or GPfates[4], see details in Supplementary Methods). We let $\mathcal{L}_l = \{i : Z_{li} = 1\}$ denote the set of cells assigned to lineage $l$. In addition, we allow the inclusion of $p$ known cell-level covariates (e.g., batch, age, or gender), represented by an $n \times p$ matrix $\mathbf{U}$, with $i$th row $\mathbf{U}_i$ corresponding to the $i$th cell, and regression parameters $\boldsymbol{\alpha}_g$ of dimension $p \times 1$. Differences in sequencing depth or capture efficiency between cells are accounted for by cell-specific offsets $N_i$.

The smoothing spline $s_{gl}$, for a given gene $g$ and lineage $l$, can be represented as a linear combination of $K$ cubic basis functions,

$$s_{gl}(t) = \sum_{k=1}^{K} b_k(t) \beta_{glk}, \qquad (2)$$

where the cubic basis functions $b_k(t)$ are enforced to be the same for all genes and lineages.

Since a separate smoothing spline is estimated for every lineage in the trajectory, we can assess DE within or between lineages by comparing the parameters $\beta_{glk}$ of these smoothing splines, see "Methods" for details. In tradeSeq, we have implemented Wald tests to assess DE and provide a range of different testing procedures that allow biologists to interpret complex trajectories in dynamic biological systems (see Fig. 2).

**Simulation study**. To benchmark relevant differential expression methods, we generated multiple data sets, spanning three distinct trajectory topologies (Fig. 3a–c), using the independently developed dynverse toolbox[1] (see "Methods"). Note that the simulated data sets are relatively "clean", as reflected by the high sensitivity and specificity of most methods. In particular, cells are approximately uniformly distributed along each lineage and often balanced between lineages. The data sets are, however, still useful to provide a relative ranking of the methods.

We demonstrate the versatility of tradeSeq by using it downstream of three trajectory inference methods, slingshot[3], Monocle 2[5], and GPfates[4], which will be denoted by tradeSeq_slingshot, tradeSeq_Monocle2, and tradeSeq_Gpfates, respectively. However, we find that GPfates fails to recover the expected trajectory topology if run in an unsupervised way (Supplementary Fig. 3a). Feeding the true pseudotimes as input to GPfates may, however, result in meaningful trajectories (Supplementary Fig. 3b). We therefore adopt this approach in the simulation study, but note that this may provide an a priori

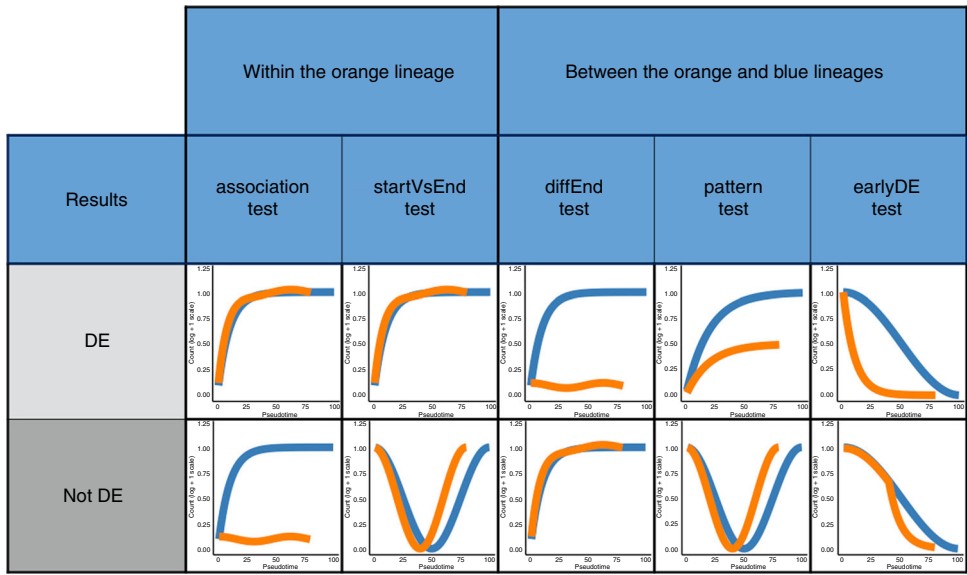

**Fig. 2 Tests currently implemented in the tradeSeq package.** Each column corresponds to a test. Tests are broken down into two categories, depending on whether they concern a within-lineage comparison, i.e., properties of the orange curve, or a between-lineage comparison, i.e., contrasting the blue and orange curves. For each test, we have two toy examples of gene expression patterns. The top one corresponds to a differentially expressed gene according to the test, while the bottom one does not.

competitive advantage to GPfates over other TI methods and that this would be impossible for real data sets.

Existing frameworks for differential expression analysis are not modular, in the sense that the DE method is tied to the TI method implemented in the same software package. Because of this, the comparison of DE methods is confounded with the quality of the upstream trajectory inference. We therefore also evaluate all trajectory-based DE methods by using the simulation ground truth as input for the DE analysis, which avoids such a confounding. GPfates was left out of this comparison, since we were not able to input the simulation ground truth to the method.

Within-lineage DE: First, we look for genes whose expression is associated with pseudotime for data sets with a cyclic topology (e.g., Fig. 3a). We compare the associationTest of a tradeSeq_slingshot analysis to the Moran's I test implemented in Monocle 3. We apply tradeSeq using 5 knots, as determined using the AIC (Supplementary Fig. 4). We only consider Monocle 3 because it is the only method that provides a test of the association between gene expression and pseudotime within a single lineage. For each TI method, we use the default/recommended dimensionality reduction method, which is PCA for slingshot and UMAP for Monocle 3.

Monocle 3, however, often fails to reconstruct the cyclic topology and instead may fit a disconnected or branching trajectory (Supplementary Fig. 5). The Moran's I test still has reasonably high sensitivity, possibly because it relies on nearest neighbors in the reduced dimensional space and not on the inferred trajectory. tradeSeq downstream of slingshot provides superior performance to discover genes whose expression is associated with pseudotime (Fig. 3d). We also compared both methods using the same dimensionality reduction input, by having slingshot infer trajectories in the UMAP space that is used by Monocle 3. The performance of tradeSeq was generally similar for both dimensionality reduction methods, except for two out of ten data sets (Supplementary Fig. 6). In all data sets, tradeSeq had better performance than Monocle 3. Finally, we evaluate an edgeR-based associationTest through fitting the NB-GAMs with edgeR instead of with mgcv (method edgeR_assoc, see "Methods" for details), and note that its performance is

similar to the tradeSeq associationTest (Supplementary Fig. 7). This could be expected because few basis functions were selected for this simulation setting. In applications that require a rich basis, however, the edgeR implementation will be prone to overfitting.

Between-lineage DE: For the bifurcating data sets (e.g., Fig. 3b), we assess differential expression between lineages using the diffEndTest and the patternTest from tradeSeq, downstream of TI methods slingshot, Monocle 2, and GPfates. We apply tradeSeq with four knots, as determined using the AIC (Supplementary Fig. 8). We compare these tests with available approaches for trajectory-based differential expression analysis, namely, BEAM (implemented in Monocle 2), GPfates, and ImpulseDE2. Furthermore, we compare against the discrete DE method edgeR, where we supervise the test to assess DE between the clusters at the true endpoints of each lineage, as derived through $k$-means clustering in PCA space. For each TI method, we use the default/recommended dimensionality reduction, which is PCA for slingshot, GPLVM for GPfates, and DDRTree for Monocle 2. For ImpulseDE2, we use the same input as for tradeSeq, i.e., derived by slingshot TI.

Monocle 2 and GPfates fail to detect the correct topology of the trajectory (i.e., a bifurcation) in, respectively, three and four out of the ten data sets (Supplementary Figs. 9 and 10). In addition, out of the remaining seven data sets, Monocle 2 misplaces the bifurcation in four of them, causing the two simulated lineages to be merged into the same lineage and creating another incorrect lineage (Supplementary Fig. 9). This strongly obscures the DE testing results. slingshot, on the other hand, correctly identifies the topology and reconstructs the trajectory for all ten data sets.

Figure 3e shows performance curves for the three data sets for which all methods are able to recover the true topology of the simulated trajectory. The tradeSeq patternTest has superior performance regardless of the TI method. Only edgeR achieves a similar performance. This is not surprising since the edgeR analysis is supervised to compare the true cell populations at the endpoints of the lineages. Interestingly, tradeSeq's diffEndTest based on the slingshot trajectory performs comparably with a supervised edgeR analysis. This is especially encouraging, since

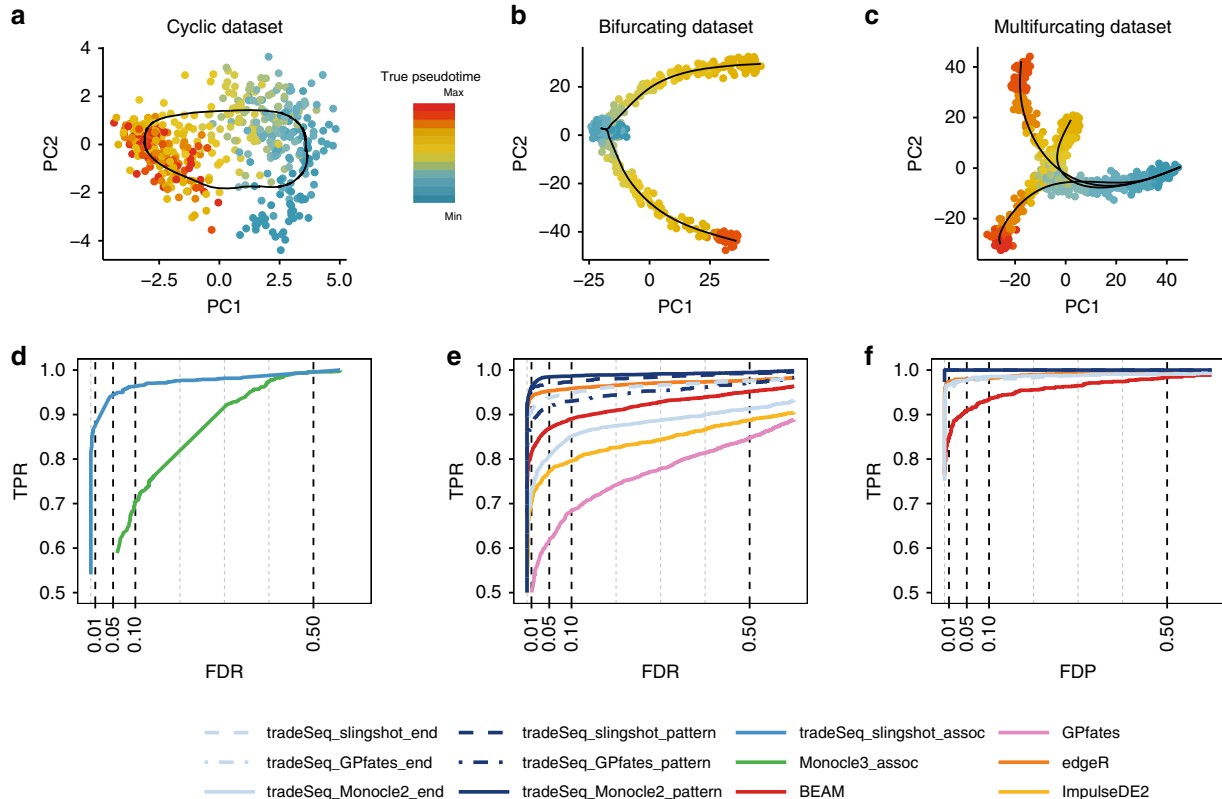

**Fig. 3 Simulation study results.** PCA plots for the (**a**) cyclic, (**b**) bifurcating, and (**c**) multifurcating simulated trajectories. The plotting symbol for each cell is colored according to its true pseudotime; trajectories (in black) were inferred by princurve in (**a**) and slingshot in (**b**) and (**c**). **d–f** Scatterplot of the true positive rate (TPR) vs. the false discovery rate (FDR) or false discovery proportion (FDP) for various DE methods applied to the simulated data sets. Panel (**d**) displays the average performance curves of DE methods across seven out of ten cyclic data sets for which all DE methods worked (Monocle 3 errored on three data sets). The `associationTest` from tradeSeq has superior performance for discovering genes whose expression is associated with pseudotime, as compared with Monocle 3. When investigating differential expression between lineages of a trajectory, the `patternTest` of tradeSeq consistently outperforms the `diffEndTest` across all three TI methods, since it is capable of comparing expression across entire lineages. Panel (**e**) displays the average performance curves across the three bifurcating data sets where all TI methods recovered the correct topology. Here, all tradeSeq `patternTest` workflows, tradeSeq_slingshot_end, and edgeR have similar performance and all are superior to BEAM, ImpulseDE2, and GPfates. Note that the performance of tradeSeq_Monocle2_end deteriorates as compared with tradeSeq_slingshot_end; the curve for tradeSeq_GPfates_end is not visible in this panel due to its low performance. For the multifurcating data set of panel (**f**), tradeSeq_slingshot has the highest performance, closely followed by tradeSeq_Monocle2 and edgeR.

the `diffEndTest` is a smoother-based analog of discrete DE. For TI methods Monocle 2 and GPfates, `diffEndTest` performs poorly, which is not surprising since the endpoints are typically ill-defined or artificially extended in the inferred trajectories for these methods (Supplementary Figs. 9 and 10). In general, BEAM, ImpulseDE2, and GPfates are outperformed by the other methods. Across all methods, tradeSeq_slingshot has the best performance. Finally, we recapitulate that the performance curves in Fig. 3e do not provide a complete view of method performance, since seven out of ten data sets were not used because at least one method failed to recover the simulated trajectory. Supplementary Fig. 11 shows mean performance curves across all ten data sets for all methods, which clearly demonstrates the superiority of tradeSeq as a DE method and of slingshot as an upstream TI method. The performance and trajectories for all ten individual data sets are shown in Supplementary Fig. 12.

In order to avoid the comparison of DE methods being obscured by differences in the upstream dimensionality reduction and trajectory inference methods, we compared tradeSeq, BEAM, and ImpulseDE2 on the simulation ground truth. We fit the tradeSeq NB-GAM once with three knots, for comparability with the BEAM approach that also uses three knots, and once with

four knots, which was found to be optimal according to the AIC (Supplementary Fig. 8). The tradeSeq `patternTest` is unaffected by the change in the number of knots and outperforms all other methods for differential expression analysis (Supplementary Fig. 13). The performance of the tradeSeq `diffEndTest` is somewhat sensitive to the number of knots, but still better than that of ImpulseDE2 and BEAM. Generally, ImpulseDE2 performs better than the BEAM approach.

For the multifurcating data set, we forego a comparison with GPfates, since it is restricted to discovering only a single bifurcation (Supplementary Fig. 14). We fit tradeSeq with three knots, as determined using the AIC (Supplementary Fig. 15). The `patternTest` from tradeSeq_slingshot and tradeSeq_Monocle2 have highest performance, closely followed by edgeR and the `diffEndTest` for these respective TI methods (Fig. 3f). BEAM was found to have the lowest performance.

Taken together, these results suggest that tradeSeq is a powerful and flexible procedure for assessing DE along and between lineages. Although tradeSeq is modular and can be used downstream of any TI method that provides pseudotime estimates, the choice of dimensionality reduction and TI method is crucial for the performance of the downstream analysis. The best performance was found for a tradeSeq_slingshot analysis, so

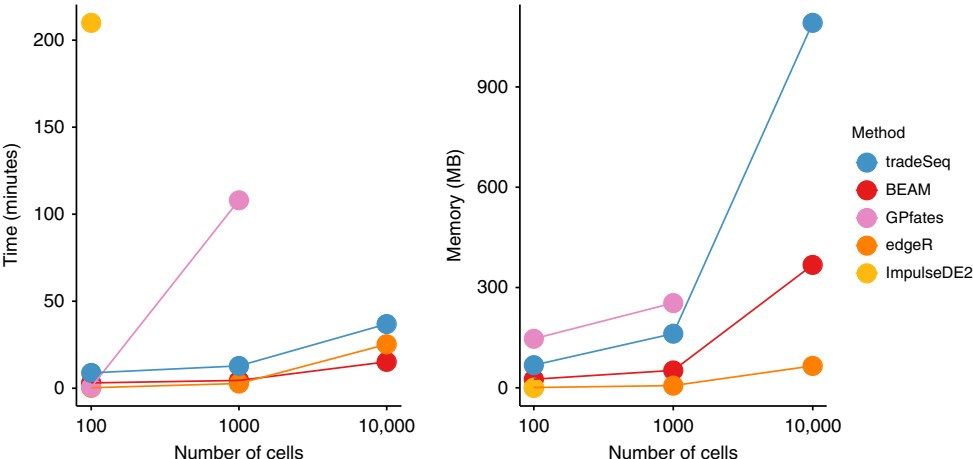

**Fig. 4 Benchmark of computation time and memory usage.** Data sets with 100, 1000, and 10,000 cells are simulated and each method is evaluated, respectively, 10, 2, and 2 times on each data set to assess computation time and memory usage. The average across iterations is plotted for each method. Methods that went over a 4-h mark were stopped and deemed taking too long.

we will mainly focus on slingshot as TI method for the real data sets.

**Computation time and memory-usage benchmark**. To assess time and memory requirements, scRNA-seq data sets with a bifurcating trajectory were simulated using the same framework as in the simulation study; the results are shown in Fig. 4, and more extensively described in Supplementary Note 1. Briefly, ImpulseDE2 is by far the slowest, taking over 3.5 h to run on a small data set of 100 cells. GPfates runs fast ( ~30 s) on the small data set, but scales poorly. BEAM, edgeR, and tradeSeq are quite fast and scale very well, even to large data sets, with BEAM scaling the best. In terms of memory requirements, all methods scale well to 10,000 cells.

**Case studies**. We analyze four case study data sets with tradeSeq: a bulk RNA-seq time-course and scRNA-seq MARS-seq, Smart-Seq, and 10× data sets. While we discuss the MARS-seq and Smart-seq data sets in the main paper and their corresponding Supplementary Notes (see below), we only report the results for the bulk RNA-seq time-course and 10× data sets in Supplementary Notes 2 and 3, respectively.

**Mouse bone marrow data set**. Paul et al.[15] study the evolution of gene expression for myeloid progenitors in mouse bone marrow. They construct a reference compendium of marker genes that are indicative of development from myeloid multipotent progenitors to erythrocytes and several types of leukocytes.

In order to compare our approach with BEAM, we are restricted to the dimensionality reduction procedures implemented in Monocle 2. We therefore first used ICA as dimensionality reduction method (Fig. 5a) in the "Discovering cell type markers" paragraph, but observed that this approach does not fully preserve the underlying biology (Supplementary Note 4). In subsequent sections, we will therefore demonstrate the powerful interpretation of a tradeSeq_slingshot analysis based on UMAP dimensionality reduction (Fig. 5b). This additionally illustrates the flexibility of tradeSeq (and slingshot) to be applied downstream of any dimensionality reduction method.

In this case study, we apply tradeSeq with six knots, as found to be optimal by the AIC (Supplementary Fig. 22). We first identify marker genes for the progenitor and differentiated cell types in the "Discovering cell type markers" paragraph. Next, we assess

which genes behave differently along the two lineages in the "Discovering progenitor population markers" paragraph. Finally, we demonstrate how one can group genes in clusters that share similar expression patterns in the "Gene expression families" paragraph.

Discovering cell type markers: tradeSeq provides the flexibility to test several interesting and distinct hypotheses for this data set, that cannot always be considered with other methods. For instance, we can find marker genes for the progenitor cell population vs. the differentiated leukocytes or erythrocytes with the `startVsEndTest` procedure (results shown in Supplementary Fig. 23). By contrasting the endpoints of the smoothers with the `diffEndTest` procedure, i.e., comparing the differentiated leukocyte and erythrocyte cells themselves, we can also discover marker genes for the differentiated cell types. For the latter, tradeSeq finds 2233 significantly differentially expressed genes at a 5% nominal FDR level, while BEAM discovers 584 genes at a 5% nominal FDR level when testing whether the association between gene expression and pseudotime depends on the lineage (Benjamini-Hochberg FDR-controlling procedure[16]). In Supplementary Note 4, we confirm that tradeSeq provides relevant biological results as compared with both BEAM and a cluster (cell type)-based comparison with edgeR. tradeSeq can thus provide relevant biological results without using the cell-type labels. Moreover, while cluster-based comparisons can be powerful in some cases, many hypotheses are difficult to assess with discrete DE, as demonstrated in the following paragraphs.

Discovering progenitor population markers: In addition to looking for markers at the differentiated cell-type level, we could also look for markers of developing myeloid cells. tradeSeq's `patternTest` can accommodate this by identifying genes with significantly different expression patterns between lineages. Remarkably, the top six genes (*Mpo*, *Prtn3*, *Ctsg*, *Car2*, *Elane*, and *Srgn*, Fig. 5c) are all confirmed as biomarkers in the extensive analysis of the original manuscript of Paul et al.[15], confirming the relevant ranking of `patternTest`. Indeed, *Prtn3* was found to be monocyte-specific, while *Mpo* and *Car2* discriminated between erythroid lineage progenitors and myeloid lineage progenitors. The cluster of genes *Elane*, *Prtn3*, and *Mpo* were the strongest markers for myeloid lineage progenitors and monocytes. In summary, all six top genes were labeled as "key genes" for hematopoiesis[15].

It might also be interesting to examine genes with significantly different expression patterns between lineages, that show little

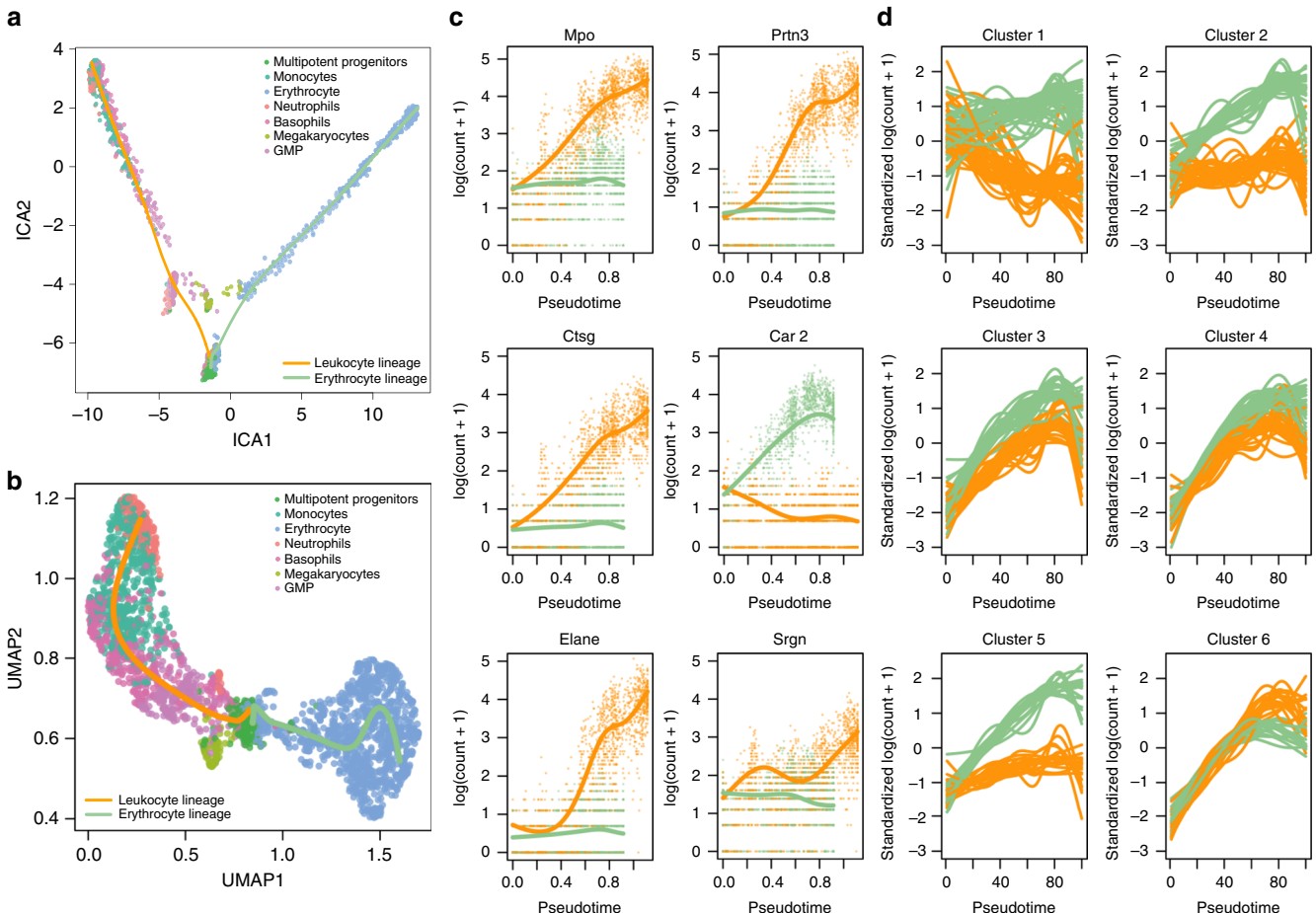

**Fig. 5 Mouse bone marrow case study. a** Two-dimensional representation of a subset of the data using independent components analysis (ICA). The myeloid trajectory inferred by slingshot is displayed. **b** Two-dimensional representation of a subset of the data using UMAP. The myeloid trajectory inferred by slingshot is displayed. The UMAP dimensionality reduction method better captures the smooth differentiation process than ICA. **c** Estimated smoothers for the top six genes identified by the tradeSeq `patternTest` procedure on the trajectory from (**b**). **d** Six clusters for the top 500 genes with different expression patterns between the two lineages (as identified by `patternTest` from tradeSeq).

evidence for DE at the endpoints. In Supplementary Note 4, we show how the combination of the results from `patternTest` and `diffEndTest` yields highly informative genes showing transient expression differences between the lineages. Note that this analysis is not possible with any other method available, since these only test for global differential gene expression between lineages.

Gene expression families: Modeling gene expression in terms of smooth functions of pseudotime opens the door for additional downstream interpretation of results that are impossible with discrete DE methods, such as the clustering of genes based on their fitted expression patterns. In general, we found that RSEC clustering provides a more stable clustering than partitioning around medoids (PAM) (Supplementary Fig. 25), the latter of which is also used by Monocle to cluster genes. For example, we can cluster the expression patterns for genes that were deemed significant by tradeSeq's `patternTest` (see "Methods", section "Clustering gene expression patterns"). This identifies gene families that have similar expression patterns within every lineage, and also similar fold changes between the two lineages (Fig. 5d shows six clusters). These gene sets can then be further screened for interesting patterns and validated by the biologist. Note that, for instance, the expression smoothers can be used to assess specific transient changes in expression during development, the signal for which might be diluted in cluster-based DE.

**Mouse olfactory epithelium data set.** Fletcher et al.[17] study the development of horizontal basal cells (HBC) in the olfactory epithelium (OE) of mice. They activate the HBCs to be primed for development, which subsequently give rise to three different cell types: sustentacular cells, microvillous cells, and olfactory sensory neurons (Fig. 6a, b). The olfactory sensory neurons are connected to the olfactory bulb for signal transduction of smell, and the sustentacular cells are general supportive cells in the OE. The function of microvillous cells, however, is not well understood; while some cells have axons ranging to the olfactory bulb, potentially indicating a sensory neuron function, others lack a basal process or axon[18]. The samples from Fletcher et al.[17] were processed using the Fluidigm C1 system with SMART-Seq library preparation, hence we expect zero inflation to be present in this data set. We therefore fit ZINB-GAMs to analyze the data using tradeSeq downstream of slingshot. Zero inflation weights are estimated with the ZINB-WaVE method[19], using the cluster labels and batch as covariates. We fit tradeSeq with six knots, as determined using the AIC (Supplementary Fig. 26). We were unable to fit a model for 0.8% of all 14,261 genes due to convergence issues of the ZINB-GAM. Note that currently no other trajectory-based DE method can account for zero inflation or provide the range of tests available in tradeSeq; hence, we forgo a comparison with other methods aside from a ZINB-edgeR analysis[14].

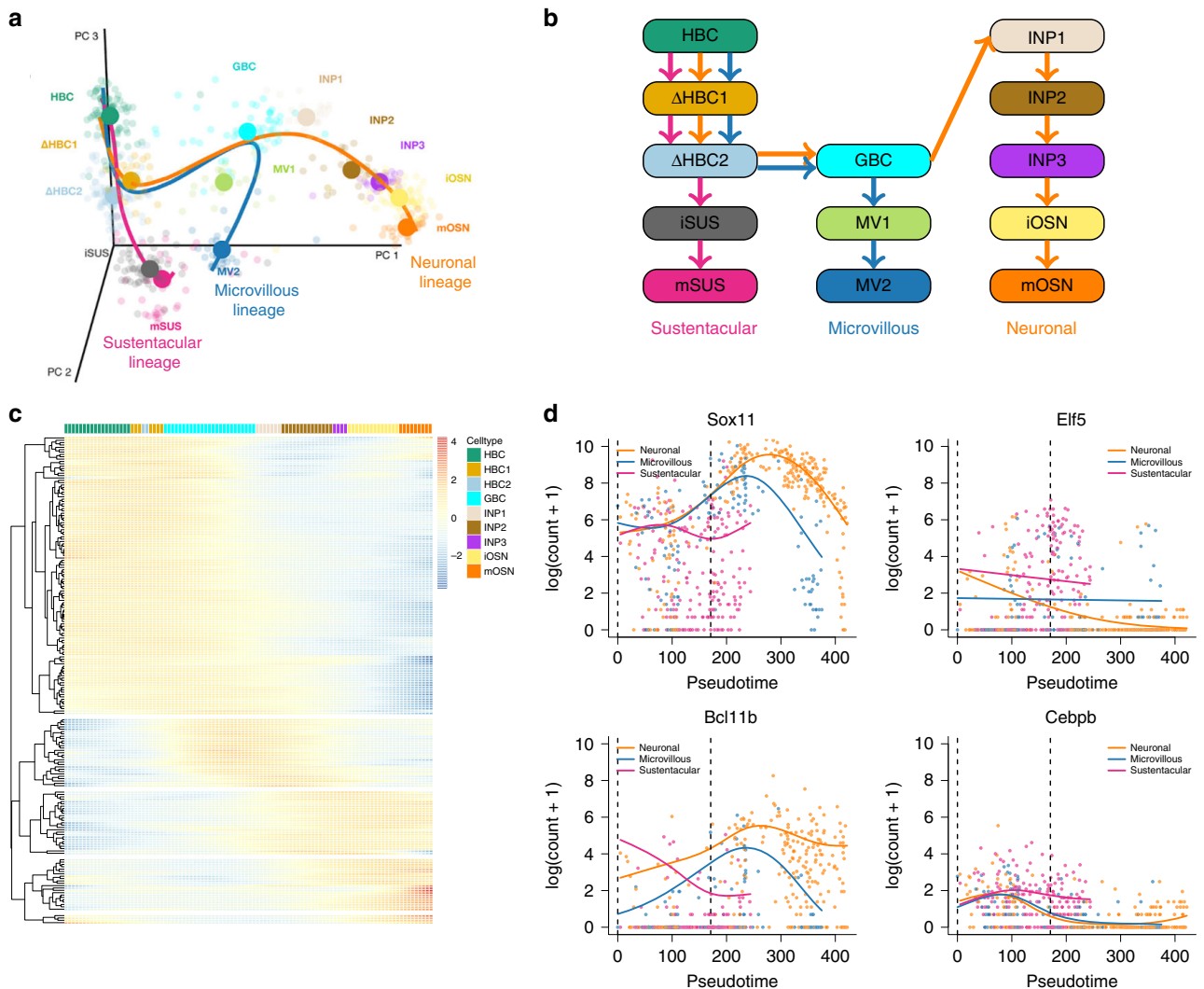

**Fig. 6 Mouse olfactory epithelium case study. a** Three-dimensional PCA plot of the scRNA-seq data, where cells are colored according to their cluster membership as defined in the original paper (see "Methods"). The simultaneous principal curves for the lineages inferred by slingshot are displayed. This Figure is reprinted from Fletcher et al.[17] with permission from the publisher. **b** Schematic of the cell types and their ordering along the lineages. **c** Heatmap for the top 200 genes that are associated with the neuronal lineage, as identified with the `associationTest` procedure from tradeSeq. Five clear gene clusters can be identified, each with a different region of activity during the developmental process. **d** Four transcription factors discovered by the `earlyDETest` between the pseudotimes of knots one and three (knots indicated with vertical dashed lines) and that are involved in epithelial cell differentiation.

In this case study, we first consider differential expression within each lineage in the "Within-lineage DE" paragraph, after which we assess differences between the three developmental lineages in the "Between-lineage DE" paragraph.

Within-lineage DE: We first consider differential expression along the neuronal lineage (the orange lineage in Fig. 6a). Using the `associationTest` implemented in tradeSeq, we recover 2730 genes at a 5% nominal FDR level. Within the top DE genes, clear clusters of expression can be observed (Fig. 6c), that are more active either at the beginning of the lineage, at specific locations along the lineage, or at the end of the lineage. Since Fletcher et al.[17] observed that cells associated with the neuronal lineage undergo mitotic division during differentiation, we investigate whether we can recover the cell cycle biology using the `associationTest`. Indeed, many of the top genes are related to the cell cycle (Supplementary Fig. 27).

We also seek biological markers that differentiate the progenitor cells from the differentiated cell types in any of the three lineages using the `startVsEndTest` procedure as part of

a global test (i.e., gene expression is compared between the start and end states for each lineage and the evidence is aggregated across the three lineages using a global test; see "Methods") and then look for enriched gene sets for the top 250 genes. The results for the top 20 gene sets (Supplementary Table 1) clearly reflect the biology of the experiment (Supplementary Note 5).

Between-lineage DE: Next, we compare the three lineages by assessing differences in their expression patterns through stage-wise testing with the `patternTest` procedure (see "Methods"). At the screening stage, we first test whether any two lineages have significantly different expression patterns. The genes that pass the screening stage are then further assessed to discover which specific pairs of lineages are deviating in their expression pattern. The screening stage identifies 3275 genes that have different expression patterns between any pair of lineages, at a 5% nominal FDR level (as reference, the top six genes are plotted in Supplementary Fig. 28). As could be expected, a large majority of the genes (2481) are significant in the neuronal–sustentacular lineage comparison. However, remarkably, we discover more DE

genes when comparing the microvillous and neuronal lineages (2149 genes) than when comparing the microvillous and sustentacular lineages (1374 genes), even though the microvillous lineage shares a longer path with the neuronal lineage. Out of all significant genes, 827 genes were identified in all three pairwise comparisons. Investigating the top 20 enriched gene sets based on the MSigDB database reveals that 12/20 of the top gene sets are related to the mitotic cell cycle (Supplementary Table 2). This is reassuring, since only the neuronal and microvillous lineages go through the cell cycle, according to Fletcher et al.[17]. In addition, we find gene sets related to neurogenesis, referring to the development of olfactory sensory neurons. The functional interpretation of the results from the combined ZINB and tradeSeq analysis hence confirms the biology of the experiment and the battery of possible tests unlock a more detailed and meaningful interpretation of the results.

None of the previously developed trajectory-based methods for assessing differential expression between lineages can currently accommodate zero inflation. In Supplementary Note 5, we compare the ZINB-tradeSeq analysis with a ZINB-edgeR analysis, and demonstrate the relevance of the genes uniquely found by tradeSeq. In addition, we illustrate the functionality of the `earlyDETest` to identify genes that may drive the differentiation around the first branching point.

## Discussion

We have proposed tradeSeq, a novel suite of tests for identifying dynamic temporal gene regulation using single-cell RNA-seq data. These tests allow researchers to investigate a range of hypotheses related to temporal gene expression, ranging from the general to the highly specific. Whereas previous methods only provide global tests of differential expression along or between lineages, tradeSeq offers a highly flexible framework that can be adapted to a single lineage, multiple lineages, or specific points or ranges along lineages. The flexibility provided by tradeSeq is crucial, as trajectory-based DE is often the final (or near final) step in a much longer analysis pipeline.

Our analyses are based on the NB-GAM of Eq. (1), which conditions on cell pseudotimes and hence ignores the fact that pseudotimes are typically inferred random variables. We therefore expect some uncertainty in pseudotime values, which may or may not be quantified by a particular TI method. Even when measures of pseudotime variability are available, neither tradeSeq nor other methods such as BEAM and GPfates currently make use of this information. Instead, all of these methods treat the pseudotimes as fixed and known. The BranchedGP method allows for uncertainty in the assignment of cells to lineages and relies on branching Gaussian processes to identify gene-specific branching dynamics[20]. However, it is computationally very intensive, with reported computation time of 2 min per gene on a data set that has been subsampled to 467 cells[20]; we therefore did not consider this method in our evaluation.

While we generally assume that pseudotime values are on similar scales across lineages, this may not always be the case. Furthermore, Trapnell et al.[11] noted that any trajectory inference method can produce pseudotime values that are not necessarily reflective of true biological time. At best, pseudotime values represent some monotonic transformation of the true maturity of each cell. Therefore, some authors have proposed the use of dynamic time warping to align pseudotime values from different experiments on potentially different scales[21]. This approach can be beneficial in cases where, for example, one lineage is much longer or shorter than another. If a gene, in reality, has a similar pattern of expression along two such lineages, this pattern could, for instance, consume 75% of the shorter lineage, but only 25% of

the longer lineage. As such, the gene could be called DE by the `patternTest` procedure. However, applying the same test after dynamic time warping may yield a negative result. Since tradeSeq only requires the estimated pseudotimes as input, which could be warped or not, it is compatible with any form of warping between lineages. We urge users to carefully consider whether pseudotime values across lineages are comparable and, if not, consider such warping strategies before comparing patterns of expression with tradeSeq.

Moving forward, it may be possible to fit ZINB-GAMs in a single step by numerically maximizing the ZINB-GAM likelihood. This could improve upon the two-step approach that we have taken in this paper, where (i) posterior probabilities of zero inflation are first estimated using ZINB-WaVE and (ii) subsequently used to unlock the NB-GAM for DE analysis in the presence of excess zeros.

In this paper, we have demonstrated tradeSeq on several scRNA-seq data sets. However, the tests that we provide downstream of the `fitGAM` function are applicable beyond this setting. Indeed, the framework may also be applicable to, e.g., downstream analysis of chromatin accessibility trajectories in scATAC-seq data sets (e.g., Chen et al.[22]) or bulk RNA-seq time-course studies; we have demonstrated the latter in Supplementary Note 2.

While we propose a number of tests based on the NB-GAM, it is important to realize that users may also implement their own statistical tests related to their specific hypotheses of interest. We therefore welcome contributions of new tests to the GitHub repository (https://github.com/statOmics/tradeSeq) of the package.

Single-cell RNA-seq tends to produce noisy data requiring long analysis pipelines in order to glean biological insight. While "all-in-one" tools that simplify this analysis may be attractive from a user's standpoint, they are not guaranteed to offer the best methods for each individual step. We therefore propose a more modular approach that expands upon previous work and opens up new classes of questions to be asked and hypotheses to be tested.

## Methods

**Negative binomial generalized additive model.** We build on the generalized additive model (GAM) methodology to model gene expression profiles as non-linear functions of pseudotime for the different lineages in a complex trajectory. In our GAM framework, each lineage is represented by a separate cubic smoothing spline, i.e., a linear combination of cubic basis functions of pseudotime. The flexibility of GAM also allows us to easily adjust for other covariates or confounders such as treatment and batch. The discrete nature and the overdispersion of read counts is addressed by modeling the expression measures $Y_{gi}$, for a given gene $g \in \{1, \dots, G\}$ across cells $i \in \{1, \dots, n\}$, using a negative binomial (NB) distribution with cell and gene-specific means $\mu_{gi}$ and gene-specific dispersion parameters $\phi_g$. Hence, we propose a gene-wise negative binomial generalized additive model (NB-GAM), represented by Eq. (1) (see Results section), where the mean $\mu_{gi}$ of the NB distribution is linked to the additive predictor $\eta_{gi}$ using a logarithmic link function. The gene-wise additive predictor consists of lineage-specific smoothing splines $s_{gl}$, that are functions of pseudotime $T_{li}$, for lineages $l \in \{1, \dots, L\}$. The binary matrix $\mathbf{Z} = (Z_{li} \in \{0, 1\}: l \in \{1, \dots, L\}, i \in \{1, \dots, n\})$ assigns every cell to a particular lineage based on user-supplied weights (e.g., from slingshot[3] or GPfates[4], see details in Supplementary Methods). We let $\mathcal{L}_l = \{i : Z_{li} = 1\}$ denote the set of cells assigned to lineage $l$. In addition, we allow the inclusion of $p$ known cell-level covariates (e.g., batch, age, or gender), represented by an $n \times p$ matrix $\mathbf{U}$, with $i$th row $\mathbf{U}_i$ corresponding to the $i$th cell, and regression parameters $\boldsymbol{\alpha}_g$ of dimension $p \times 1$. Differences in sequencing depth or capture efficiency between cells are accounted for by cell-specific offsets $N_i$.

The smoothing spline $s_{gl}$, for a given gene $g$ and lineage $l$, can be represented as a linear combination of $K$ cubic basis functions (Eq. (2), see Results section), where the cubic basis functions $b_k(t)$ are enforced to be the same for all genes and lineages. Our default computational implementation sets $K = 6$. Thus, for each gene and each lineage in the trajectory, we estimate $K = 6$ regression coefficients $\beta_{glk}$. The number of parameters in the gene-wise model is $L \times K + p + 1$, which is typically much lower than the number of cells $n$ in the data set.

The NB-GAM is fitted gene by gene using the `fitGAM` function from the tradeSeq package, which relies on the mgcv package in R. We build upon recent developments in mgcv that allow the joint estimation of the NB regression parameters in $\mu_{gi}$ and dispersion parameter $\phi_g$[23]. In order to control the

smoothness of the spline, the coefficients $\beta_{glk}$ are shrunken by substracting a penalty $\lambda_g \beta_g^T \mathbf{S} \beta_g$ from the log-likelihood function, where $\beta_g$ denotes the concatenation of the $LK$-dimensional column vectors $\beta_{gl}$ of lineage-specific smoother coefficients and $\mathbf{S}$ is an $(LK) \times (LK)$ diagonal matrix that indicates which coefficients in $\beta_g$ are to be penalized. The magnitude of penalization is controlled by the smoothing parameter $\lambda_g$, which is selected using generalized cross-validation[24]. Note that we enforce identical basis functions between lineages, i.e., $b_k$ does not depend on $l$, as well as identical smoothing parameter $\lambda_g$, in order to ensure that the smoothers are comparable across lineages.

Importantly, the model of Eq. (1) can accommodate zero-inflated counts typical for full-length scRNA-seq protocols by using observation-level (i.e., cell-level) weights obtained, for instance, from the zero-inflated negative binomial (ZINB) approach of Van den Berge et al.[14] and Risso et al.[19].

**Choosing an appropriate number of knots**. Ideally, the number of knots $K$ should be selected to reach an optimal bias-variance trade-off for the smoother, where one explains as much variability in the expression data as possible with only a few regression coefficients (see Supplementary Fig. 1). In practice, the number of knots $K$ may be selected by evaluating the Akaike information criterion (AIC) using the evaluateK function implemented in tradeSeq. We have deliberately chosen the AIC as evaluation criterion, since the Bayesian information criterion (BIC) seemed to favor overly complex models (i.e., an excessively high number of knots). The knots are by default positioned according to the quantiles of the pseudotime values. For example, if a smoother is fit with three knots, then there will be a knot at the minimum, median, and maximum pseudotime values. The knots may be interpreted as relative markers of progress along the trajectory. However, it is important to realize that this might not necessarily linearly correlate with true chronological time.

**Statistical inference**. We propose a general and flexible testing framework for (linear combinations of) the parameters $\beta_g$, which allows us to pinpoint specific types of differences in gene expression both within and between lineages; see Fig. 1 for an overview. We first present the general approach and then detail the implementation and interpretation of specific DE tests.

All proposed DE procedures involve testing null hypotheses of the form $H_0: \mathbf{C}^T \beta_g = 0$ using Wald test statistics

$$W_g = \hat{\beta}_g^T \mathbf{C} (\mathbf{C}^T \hat{\Sigma}_{\hat{\beta}_g} \mathbf{C})^{-1} \mathbf{C}^T \hat{\beta}_g, \qquad (3)$$

where $\hat{\beta}_g$ denotes an estimator of $\beta_g$, $\hat{\Sigma}_{\hat{\beta}_g}$ represents an estimator of the covariance matrix $\Sigma_{\hat{\beta}_g}$ of $\hat{\beta}_g$, and $\mathbf{C}$ is an $(LK) \times C$ matrix representing the $C$ contrasts of interest for the DE test.

For each gene, we compute $p$-values based on the nominal chi-squared asymptotic null distribution of the Wald statistics (with degrees of freedom equal to the column rank of $\mathbf{C}$). Rather than attaching strong probabilistic interpretations to the $p$-values (which, as in most RNA-seq applications, would involve a variety of hard-to-verify assumptions and would not necessarily add much value to the analysis), we view the $p$-values simply as useful numerical summaries for ranking the genes for further inspection. There are five tests currently implemented in the tradeSeq package, which are introduced in detail in the sections below. Fig. 2 provides a visual overview of the scope of each test.

**Within-lineage comparison tests**. associationTest: A relevant first question is whether gene expression is associated with pseudotime along a given lineage, i.e., whether the smoother is flat or varying along pseudotime. To address this question, the associationTest tests the null hypothesis that all smoother coefficients within the lineage are equal, i.e., $H_0: \beta_{glk} = \beta_{glk'}$ for all $k \neq k' \in \{1, \ldots, K\}$. This null hypothesis can be encoded in several ways; here, we chose the contrast matrix $\mathbf{C}$ to be an $LK \times L (K-1)$ matrix, where each column corresponds to a contrast between two consecutive $\beta_{glk}$ and $\beta_{gl(k+1)}$ and where we have $K-1$ contrasts per lineage for a total of $L (K-1)$ contrasts.

startVsEndTest: By default, the startVsEndTest compares mean expression at the progenitor state (i.e., the start of the lineage) to mean expression at the differentiated state (i.e., the end of the lineage). Specifically, $\mathbf{C}$ is an $(LK) \times L$ matrix, whose entry in row $k + (l-1)K$ and column $l$ encodes the contrast for lineage $l$ and knot $k$ and is defined by $b_k(T_{l,\max}) - b_k(T_{l,\min})$, where $T_{l,\max} = \max_{\{i:i \in \mathcal{L}_l\}} T_{li}$ and $T_{l,\min} = \min_{\{i:i \in \mathcal{L}_l\}} T_{li}$ denote, respectively, the maximum and minimum pseudotime across all cells assigned to lineage $l$. Other entries of $\mathbf{C}$ are set to zero. Therefore, the $l^{\text{th}}$ element of the vector $\mathbf{C}^T \beta_g$ is $\sum_{k=1}^K (b_k(T_{l,\max}) - b_k(T_{l,\min})) \beta_{glk} = s_{gl}(T_{l,\max}) - s_{gl}(T_{l,\min})$, which contrasts mean expression at the beginning and at the end of the lineage. Note that contrasting the start and endpoints of a lineage is a special case of a more general capability of tradeSeq to compare the mean expression between any two regions of a given lineage. As such, this test can be considered a generalization of cluster-based discrete DE within a lineage (e.g., Risso et al.[25]).

**Between-lineage comparison tests**. diffEndTest: The diffEndTest compares average expression at the differentiated states of multiple lineages, i.e., it compares the endpoints of different lineage-specific smoothers. It can be viewed as an analog of discrete DE for the differentiated cell types. The test is implemented using a Wald test statistic, as described above, where $\mathbf{C}$ is an $(LK) \times L (L-1)/2$ matrix. Each column of $\mathbf{C}$ encodes a pairwise contrast between the endpoints of two lineages, such that the corresponding element of $\mathbf{C}^T \beta_g$ is $s_{gl_1}(T_{l_1,\max}) - s_{gl_2}(T_{l_2,\max})$ for lineages $l_1$ and $l_2$.

patternTest: This test compares the expression patterns along pseudotime between lineages by contrasting a fixed set of equally spaced pseudotimes ($M = 100$ by default). First selecting the pseudotimes and subsequently comparing their expression levels between lineages, allows for comparisons between smoothers of different lengths. Specifically, for lineage $l$, let $P_{lm}$ denote the $m$th equally spaced pseudotime between $T_{l,\min}$ and $T_{l,\max}$. The contrast of $M$ points corresponds to testing the null hypothesis that a gene has the same expression pattern along pseudotime across the lineages under comparison, while normalizing for the length of the lineages. The test is implemented using a Wald test statistic, as described above, where $\mathbf{C}$ is an $(LK) \times L (L-1)M/2$ matrix. Each column of $\mathbf{C}$ encodes a pairwise comparison between two pseudotimes of two different lineages, such that the corresponding element of $\mathbf{C}^T \beta_g$ is $s_{gl_1}(P_{l_1 m}) - s_{gl_2}(P_{l_2 m})$ for lineages $l_1$ and $l_2$ and $m \in \{1, \ldots, M\}$. The test is implemented through the eigendecomposition of the estimated variance–covariance matrix of the contrasts to avoid singularity problems[26] (see Supplementary Methods). It should be noted that this test is a general test, able to identify both differences in patterns of expression as well as genes with similar patterns but different mean expression across the pseudotime range. It is therefore most useful as a screening test to identify any form of differential expression between the lineages.

earlyDETest: The earlyDETest aims to identify genes that are differentiating around a branching of the trajectory. It is similar to the patternTest, in that it also compares the expression patterns along pseudotime between lineages by contrasting a fixed set of equally spaced pseudotimes ($M = 100$ by default). However, instead of using points distributed from the beginning $T_{l,\min}$ to the end $T_{l,\max}$ of the lineages as in the patternTest, it relies on points over a shorter range of time. In the current implementation, this range is delimited by the pseudotimes of two user-specified knots. The knots should be chosen to enclose the branching event (or any event of interest) and do not need to be consecutive.

**Global testing**. While the statistical tests introduced above can assess DE within one lineage or between a pair of lineages, one may want to investigate multiple (i.e., more than two) lineages. For example, if a trajectory consists of three lineages, one may wish to test the global null hypothesis that, for each of the three lineages, there is no association between gene expression and pseudotime using the associationTest. The null hypothesis that would be tested can be expressed as $H_0: \forall l$ and $\forall k \neq k'$, $\beta_{glk} = \beta_{glk'}$, i.e., within each of the three lineages, all $K$ regression coefficients are equal. We refer to such a test as a "global test". The tradeSeq package provides functionality for global testing for each of the within and between-lineage tests described above. For within-lineage tests, the user can specify whether the test should be done for each lineage individually or at the global level (i.e., for all lineages). For between-lineage tests, the user can specify if a global test should be performed or whether all pairwise comparisons should be performed.

**Stage-wise testing**. For the mouse olfactory epithelium case study[17], we apply stage-wise testing, as implemented in stageR[27,28], to assess DE between lineages using multiple tests for each gene. Stage-wise testing aims to control the overall false discovery rate (OFDR)[27], i.e., the expected proportion of genes with at least one falsely rejected null hypothesis among all genes declared DE. In our case, the OFDR can be interpreted as a gene-level FDR[28]. Stage-wise testing is performed in two stages, a screening and a confirmation stage. At the screening stage, each gene is screened by performing a global test across all null hypotheses of interest, essentially testing whether at least one of these hypotheses can be rejected. At that stage, the FDR is controlled across genes at level $\alpha_I$. At the confirmation stage, each specific hypothesis is assessed, but only for the genes that have passed the screening stage. For each gene, the family-wise error rate (FWER) is controlled across hypotheses at level $\alpha_{II} = \frac{R}{G} \alpha_I$, where $R$ denotes the number of genes that had their global null hypothesis rejected at the screening stage and $G$ the total number of genes assessed. Heller et al.[27] proved that this procedure controls the overall FDR at level $\alpha_I$. It should be noted that, while the stage-wise testing paradigm theoretically controls the OFDR (given underlying assumptions are satisfied), the resulting $p$-values might still be too liberal since the same data are used for trajectory inference and differential expression. As mentioned before, we use $p$-values simply as numerical summaries for ranking the genes for further inspection.

**Clustering gene expression patterns**. The NB-GAM can also be used to cluster genes according to their expression patterns, as shown in Fig. 1. Specifically, for each gene, we extract a number of fitted values for each lineage (100 by default). We can then use resampling-based sequential ensemble clustering (RSEC), as implemented in clusterExperiment[25], to perform the clustering based on (the top

principal components of) the standardized fitted values matrix (i.e., the fitted values are standardized to have zero mean and unit variance across cells for each gene). Importantly, we allow for any clustering algorithm that is built-in into cluster-Experiment or chosen by the user to perform the clustering. This clustering approach is implemented in the tradeSeq package (*clusterExpressionPatterns* function) for downstream analysis facilitating the interpretation of DE genes.

**Implementation**. The above described fitting procedure, DE tests, and clustering of expression patterns are implemented in the open-source R package tradeSeq, available through the Bioconductor Project (http://www.bioconductor.org/packages/release/bioc/html/tradeSeq.html). We provide an extensive vignette along with the package, as well as a cheat sheet describing the different types of DE patterns detected with each test.

**Methods comparison**. slingshot is a fast and robust method for TI that was shown to be among the top-performing methods in a recent large-scale benchmarking study[1]. Hence, we evaluate tradeSeq downstream of a slingshot analysis, which can work with any dimensionality reduction and clustering methods. slingshot builds a cluster-based minimum spanning tree (MST) to infer the global lineage topology and make an initial assignment of cells to lineages. This structure is then smoothed by fitting simultaneous principal curves, which refine the assignment of cells to lineages. This process results in lineage-specific pseudotimes and weights of assignment for each cell.

GPfates[4] is a Python package that adopts Gaussian processes in reduced dimension to infer trajectories. Dimensionality reduction is performed using Gaussian process latent variable models (GPLVM)[29]. GPfates is able to identify bifurcation points and assess how well a bifurcation fits the expression pattern of each gene, i.e., whether the patterns of gene expression are different between the lineages. This allows us to compare a slingshot + tradeSeq analysis with a GPfates analysis. In addition, we also evaluate a tradeSeq analysis downstream of TI with GPfates, since GPfates also calculates posterior probabilities that each cell belongs to a particular lineage. We then compare the complete GPfates (TI and DE) analysis to a GPfates + tradeSeq analysis.

Monocle 2[5] applies reverse graph embedding to infer trajectories and yields a principal graph that is allowed to branch. It provides a similar approach as tradeSeq with the branch expression analysis modeling (BEAM) method. It assumes a gene-wise negative binomial model for gene expression, where the mean is expressed in terms of lineage-dependent smooth functions of pseudotime, i.e.,

$$\log(\mu_{gi}) = \sum_{l=1}^{L} (\beta_{0gl} + s_{gl}(T_{li})). \qquad (4)$$

In this model, the lineage-specific intercepts $\beta_{0gl}$ account for mean differences in expression between lineages, while the lineage-specific smoothers $s_{gl}(t)$ model the expression change along pseudotime. To test for lineage-dependent expression, the full model is compared with a null model of the form

$$\log(\mu_{gi}) = \beta_{g0} + s_g(T_i)$$

using a likelihood ratio test. Thus, BEAM tests whether the smooth functions of gene expression along pseudotime are different between lineages. Importantly, BEAM is restricted to the dimensionality reduction methods that are implemented in Monocle 2, namely DDRTree[5] and Independent Components Analysis (ICA). In addition, it only provides a screening test (like the patternTest in tradeSeq), as it only allows testing for any difference in expression profiles between lineages and does not specify the exact type of divergence.

An alpha release for Monocle 3 is available online (downloaded August 30, 2018 from the https://github.com/cole-trapnell-lab/monocle-release/tree/monocle3_alpha Monocle GitHub repository) which, unlike Monocle 2, performs uniform manifold approximation and projection (UMAP)[30] dimensionality reduction upstream of the trajectory inference. In addition, Monocle 3 implements the Moran's I test to discover genes whose expression is significantly associated with pseudotime; a functionality that is unavailable in Monocle 2.

ImpulseDE2[31] also assumes a gene-wise negative binomial model for the expression counts, where the mean is expressed as a weighted combination of two sigmoid functions. This model essentially allows the estimation of three "state-specific expression values", where the transitions between the states are modeled with the two sigmoid functions. The DE method is not linked to any trajectory inference procedure since it assumes that the pseudotime for each cell is known. In this paper, we use ImpulseDE2 downstream of slingshot. Prior to the fitting, ImpulseDE2 relies on DESeq2 for normalization and estimation of the NB dispersion parameter. However, DESeq2 cannot handle genes having at least one zero count, which is common in scRNA-seq. In such a scenario, we therefore "manually" estimate size factors and dispersion parameters using the DESeq2 poscounts normalization, which was developed to deal with this issue[14,32].

edgeR[33] is a discrete differential expression method, where the groups under comparison must be defined a priori. It is therefore useful for assessing DE between, for example, annotated clusters or different treatment groups. For such comparisons, edgeR is a powerful method with high sensitivity. Note that, while edgeR was originally developed for group-based differential expression, it would be possible to incorporate the basis functions of the smoothers as continuous covariates in the model. However, no regularization would be performed on the estimation of the smoother regression coefficients, hence the model would be prone to overfitting. A similar approach was evaluated in Fischer et al.[31], where DESeq2[34] was used to fit splines by incorporating natural cubic basis functions in the linear predictor. In addition, edgeR does not provide an implementation of the DE tests in tradeSeq. Only the associationTest is readily available in edgeR by testing whether all basis function parameters are equal to zero; the other tests would require a similar development as presented in tradeSeq. Hence, while it is possible to fit smoothers by using edgeR, instead of mgcv, we emphasize that this would merely be an alternative and less general approach to fitting the NB-GAMs we propose in our paper.

**Simulation study**. The simulation study evaluates methods that (differentially) associate gene expression with pseudotime for three different trajectory topologies, i.e., a cyclic, a bifurcating, and a multifurcating trajectory. As independent evaluation, we use the extensive trajectory simulation framework dynverse that previously served for benchmarking trajectory inference methods in Saelens et al.[1]. Interested readers should refer to the original publication for details on the data simulation procedure. Data set characteristics are listed in Table 1.

For each of the cyclic and bifurcating topologies, we generate and analyze ten data sets. Since the multifurcating topology is very variable across simulations due to its flexible definition, its analysis requires substantial supervision. Therefore, we analyze only one representative multifurcating data set.

Prior to trajectory inference, the simulated counts are normalized using full-quantile normalization[35,36]. For TI with slingshot, we apply principal component analysis (PCA) dimensionality reduction to the normalized counts and $k$-means clustering in PCA space. For the bifurcating and multifurcating trajectories, the start and end clusters of the true trajectory are provided to slingshot to aid it in inferring the trajectory. For the edgeR analysis, we assess DE between the end clusters that are also provided to slingshot. The BEAM method can only test one bifurcation point at a time. For the multifurcating data set, we therefore assessed both branching points separately and aggregated the $p$-values using Fisher's method[37]. For the tradeSeq and edgeR analyses of the multifurcating data set, we perform global tests across all three lineages.

We assess performance based on scatterplots of the true positive rate (TPR) vs. the false discovery proportion (FDP), according to the following definitions

$$\text{FDP} = \frac{FP}{\max(1, FP + TP)}$$

$$\text{TPR} = \frac{TP}{TP + FN},$$

where $FN$, $FP$, and $TP$ denote, respectively, the numbers of false negatives, false positives, and true positives. FDP-TPR curves are calculated and plotted with the Bioconductor R package iCOBRA[38].

**Table 1 Overview of simulated data sets.**

|  | Cyclic data set | Bifurcating data set | Multifurcating data set |
|---|---|---|---|
| Simulation framework | dyngen | dyntoy | dyntoy |
| Number of cells | 505–508 | 500 | 750 |
| Number of genes | 312–444 | 5000 | 5000 |
| % of DE genes | 42–47% | 20% | 20% |
| Number of lineages | 1 | 2 | 3 |
| Topology | Cyclic | Bifurcating | Multifurcating |
| Number of data sets | 10 | 10 | 1 |

Each data set is simulated using one of the frameworks from the dynverse toolbox (dyngen or dyntoy), which are designed to simulate scRNA-seq data according to trajectory topologies. Each data set can be characterized by the topology of the trajectory, as well as the number of cells and genes. Low-dimensional representations of representative data sets can be found in Fig. 3. Note that the cyclic data sets have some variation in the numbers of genes and cells and in the amount of differential expression, which is inherent to the dyngen simulation framework.

**Case studies**. Bulk RNA-seq time-course data set: As proof-of-principle case study, we analyze a bulk RNA-seq time-course data set from Kiselev et al.[39] with tradeSeq. The data were downloaded from the GitHub repository at https://github.com/daniel-spies/rna-seq_tcComp, and the original differential expression results were downloaded from https://github.com/wikiselev/rnaseq.mcf10a/tree/master/data.

Mouse bone marrow data set: We use as second case study the mouse haematopoiesis scRNA-seq data set of Paul et al.[15] Two small cell clusters corresponding to the dendritic and eosinophyl cell types were removed from the trajectory inference and downstream DE analysis, since these are outlying cell types that do not seem to belong to any particular lineage (Supplementary Fig. 2). We use the same data set as the Monocle 3 vignette, which was prefiltered to contain genes with relatively high expression. After filtering, the data set consists of 3004 genes and 2660 cells.

tradeSeq downstream of slingshot is compared with the BEAM approach from Monocle 2. Since BEAM is restricted to the dimensionality reduction methods implemented in the package, we use independent components analysis for both slingshot and Monocle 2 in this comparison. For Monocle 2, we specify the argument num_paths=2 to aid it in inferring two lineages.

Subsequently, we demonstrate a tradeSeq analysis downstream of slingshot by performing dimensionality reduction using UMAP[30], following the data processing pipeline described in the http://cole-trapnell-lab.github.io/monocle-release/monocle3/#tutorial-1-learning-trajectories-with-monocle-3 Monocle 3 vignette, since this better reflects the biology of the experiment.

In this case study, we show how one can perform multiple tests to identify genes with distinct types of behavior, specifically, genes that are deemed DE for one test (test 1), but not another (test 2). Let $W_g^{(\tau)}$ denotes the test statistic for gene $g$ in test $\tau \in \{1, 2\}$ and $\mathrm{rk}_g^{(\tau)}$ denotes the rank (in terms of ordering from low to high) of $W_g^{(\tau)}$ among all $G$ test statistics associated with the $G$ genes. Then, define a score for each gene $g$ as $\mathrm{score}_g = (\mathrm{rk}_g^{(1)})^2 + (G - \mathrm{rk}_g^{(2)})^2$. Genes with high scores are genes which are expected to be DE for test 1, but not DE for test 2 and vice versa. This is used to identify genes that are DE with the patternTest (test 1) but not diffEndTest (test 2), i.e., genes that are transiently DE between lineages. Note that the procedure only provides a ranking of the genes and not an evaluation of statistical significance.

Mouse olfactory epithelium data set: The olfactory epithelium (OE) data set from Fletcher et al.[17] is our third case study. We use the lineages discovered in the original paper. Prior to the analysis, the data set is filtered to retain genes with reasonably high expression; we consider 14,261 genes and 616 cells for downstream analysis. In brief, counts are normalized using full-quantile normalization[35,36] followed by regression-based adjustment for quality control variables[17]. Dimensionality reduction is performed through PCA on the normalized log-transformed counts that are offset by 1 to avoid taking the log of zero, i.e., $\log (y + 1)$. Clustering is performed through $k$-means on the first 50 principal components by varying the number of clusters $k \in \{4, …, 15\}$; stable clusters are derived using clusterExperiment[25], yielding a final repertoire of 13 cell clusters. Next, slingshot is used to infer trajectories with the initial cluster chosen by known marker genes of horizontal basal cells (HBC), an adult stem cell population. A double bifurcation is discovered, with the first giving rise to sustentacular cells and two more lineages that split into microvillous cells and olfactory sensory neurons. The data were downloaded from GEO with accession number GSE95601.

Adipocyte differentiation data set: As final case study, we analyze a 10× Genomics adipocyte differentiation data set described in Merrick et al.[40]. The gene expression counts were downloaded from GEO with accession number GSE128889. We focus the analysis on the single cells collected from 12-day-old mice. The raw data set consists of 27,998 genes and 11,423 cells. We only retain genes with a count of at least 2 in at least 400 cells, and normalize the data using full-quantile normalization[35]. Since not all cells in the data set are involved in the adipocyte differentiation process, we first identify the relevant clusters of cells using the marker genes described in the original manuscript. We apply $k$-means clustering ($k = 10$) to the top eight principal components of log-transformed counts. Using this clustering, we identify the relevant clusters based on the reported markers, and subsequently apply UMAP dimensionality reduction[30,41] to the top 20 principal components for that subset of cells. The processed data set consists of 2851 genes and 8071 cells. We use slingshot[3] for trajectory inference in two-dimensional UMAP space.

## Data availability
The code to generate all simulated data sets is included in the GitHub repository of the paper at https://github.com/statOmics/tradeSeqPaper. The data for the mouse bone marrow case study were downloaded from http://trapnell-lab.gs.washington.edu/public_share/valid_subset_GSE72857_cds2.RDS. The raw data for the olfactory epithelium case study are available on GEO with accession number https://www.ncbi.nlm.nih.gov/geo/query/acc.cgi?acc=GSE95601.

## Code availability
The code to reproduce the analyses, figures, and tables in the paper is available on GitHub at https://github.com/statOmics/tradeSeqPaper. The tradeSeq open-source R

package is available through the Bioconductor Project at http://www.bioconductor.org/packages/release/bioc/html/tradeSeq.html.

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

## Acknowledgements

We thank Valentine Svensson for his help on the implementation and interpretation of the output of the GPfates method. Olivier Thas and Catalina Vallejos provided constructive feedback on a draft of the paper. K.V.d.B. is a postdoctoral fellow from the Belgian Americal Educational Foundation (BAEF) and is supported by the Research Foundation Flanders (FWO), research grants G062219N, 1S41818N, and 1246220N, and travel grant 148095. W.S. and R.C. are supported by the Research Foundation Flanders (FWO, grants 11Z4518N and 11Y6218N, respectively). Y.S. is an ISAC Marylou Ingram scholar. S.D., and H.R.d.B. are supported by a grant from the National Institutes of Health BRAIN Initiative Cell Census Program (1U19MH114830-01). L.C. is supported by the Research Foundation Flanders (FWO), grant G062219N. This research received funding from the Flemish Government under the "Onderzoeksprogramma Artificiële Intelligentie (AI) Vlaanderen" program.

## Author contributions

K.V.d.B., H.R.d.B., K.S., S.D., and L.C. conceived and designed the study. K.V.d.B. and H.R.d.B. implemented the method. K.V.d.B., H.R.d.B., and K.S. analyzed the data. W.S., R.C., and Y.S. generated the synthetic data. K.V.d.B., H.R.d.B., K.S., S.D., and L.C. wrote the paper; and W.S., R.C., and Y.S. contributed to revisions of an initial draft.

## Competing interests

The authors declare no competing interests.
