## [Peer Review File · Nature Communications]

Reviewers' Comments:

Reviewer #1:

Remarks to the Author:

The manuscript "Trajectory-based differential expression analysis" describes a statistical method and package for testing gene expression differences along a pseudotime trajectory. Trajectory reconstruction is a popular question for single-cell data and novel methods are constantly being developed, yet there is a need for a more general method of assessing significance and prioritizing genes. The approach described here is flexible and agnostic to any particular reconstruction method. Overall the manuscript is well-written with nice figures. The statistical framework used here is appropriate and shows good performance. The accompanying software package runs and would be useful to a broad audience. The results and conclusions presented are valid and well supported.

Code comments:

1. I ran the exact same code in the vignette and obtained somewhat different results—in my case I only have a single lineage, which produced different plots and results in some tests not being available. I did use the same seed. Other than this difference, the code/package seems to work very well.

Writing comments:

2. There are a lot of subsections, which made it a bit difficult to follow completely through (this could just be the sizing and bold of every section since it was hard to tell where I was sometimes). More upfront motivation for why the two case-studies are analyzed differently might help with this -- they do have completely different subsection titles.

3. The Introduction is well-written. As a suggestion, the authors might consider a few sentences on the connection to somewhat similar underlying ideas for bulk RNA-seq time-series analysis methods. For example: <https://www.ncbi.nlm.nih.gov/pubmed/26046293>, <https://www.ncbi.nlm.nih.gov/pmc/articles/PMC4155246/>, <https://bmcbioinformatics.biomedcentral.com/articles/10.1186/s12859-018-2405-x>.

Also, the single-cell method TSCAN uses a GAM to test for DE along the estimated trajectory, <https://academic.oup.com/nar/article/44/13/e117/2457590>. Obviously tradeSeq is much more extensive, but this paper should be cited.

4. The end of the Introduction has a sentence saying "one can build upon our stageR package". This should be in the Discussion or should be implemented within this new package. Did the authors mean, "we have built upon"?

5. Starting with the section "Within-lineage comparisons" which lists the different tests, it would be incredibly useful to have an early figure with an example/cartoon of each of these cases highlighting the test points on the trajectory. I see the table, which is helpful but took me a bit longer to fully grasp. If space is limited in the main text, then this would be helpful in the vignette.

6. In the Methods section on earlyDETest, it says "driving the differentiation". I suggest a different phrasing here as that phrasing also connotes the specific biological process of differentiation and may be confusing to future readers.

7. Page 10 the second paragraph on "Mouse bone marrow dataset" is confusing. It says, "We will therefore first apply ICA as dimensionality reduction method (Figure 3a) in the 'Discovering cell type markers' paragraph." It would make more sense if it instead stated that the authors "first tried ICA" and explain the problems encountered as they have done, and then only use UMAP further.

8. The package dyngen is referenced by name but no explanation is given, even just saying it is a simulation of dynamic trajectories would be helpful to future readers. I suggest the authors double-check all package/methodology names are given a somewhat descriptive comment when introduced.

Methods comments:

1. Why are different methods for enrichment used in the manuscript? Fgsea package and then the MSigDB database.

2. Why is the stage-wise testing only implemented for one case study dataset? Is it fair to

compare to other methods only using the Benjamini Hochberg FDR control? Since the manuscript explicitly states that the p-values are only for ranking, it's actually unclear the motivation for this extra step.

3. How should future readers/users think of knots? As just markers of relative timing along the trajectory? For example, Figure 4 says the DE test was done between knots 2 and 4. How to interpret this choice?

4. Are the datasets pre-filtered at all or was a GAM fit for every gene?

Figure comments:

1. Figure 1 says "knots = c(1, 2)." but it's unclear what that actually means.

2. Figure 3. How is the clustering done here? In the six clusters, I assume each gene has a green and an orange line, but how are they so consistent within a cluster? Are the data-points used to cluster pulled from both lineages?

Reviewer #2:

Remarks to the Author:

The authors describe a strategy called tradeSeq to analyze per-lineage gene expression patterns in scRNA-seq trajectory data. In particular they have identified a set of hypothesis tests to query for particular smooth within-lineage trends and cross-lineage comparisons. To account for multiple testing the authors make clever use of the theory of stage-wise testing.

To model the data weighted spline regression with negative binomial likelihood is used, this is implemented as a wrapper around the mgcv package. The strategy is similar to the strategy implemented in the Monocle package.

While the choice of model likelihood is motivated by scRNA-seq data, there is nothing in particular about the model used which relates to the predictors being "pseudotime" and lineage membership.

In this study the authors also perform a comparison between pseudotime and lineage assignment methods identifying Slingshot as performing well.

Finally, case studies of using the different hypothesis tests in tradeSeq to answer questions about transient and lineage-specific expression patterns are demonstrated on published bone marrow and olfactory epithelium data.

The interpretable significance tests are solving a severe problem in the community. However, the underlying model is not correctly compared to the alternative published methods, and the authors claim other methods can only be applied in particular scenarios of upstream analysis, which is only true on a highly superficial level.

The method presented by the authors requires as input a gene expression matrix, per-cell pseudotime, and per-cell lineage membership. The same can be used with other methods.

In Monocle's BEAM test, the significance test is performed by comparing a spline GLM which includes an interaction between the spline parameters and the lineage membership with one without interaction. It is implemented as a wrapper around VGAM similar to how tradeSeq wraps mgcv. In tradeSeq regression weights are used instead of making the interaction part of the model, but the effect will be the same. The same underlying model can be fitted in Monocle using the differentialGeneTest method (which can also use other covariates such as batch).

The authors claim edgeR can only be used for discrete tests, but edgeR supports arbitrary design matrices, including ones made with splines, equivalent to the tradeSeq model or Monocle's BEAM models.

The bifurcation test in GPFates performs a likelihood ratio calculation of two OMGP models: one with pre-fitted cell lineage memberships, and one with equal membership to each trajectory. This test could also be performed with the inputs specified for tradeSeq.

Not mentioned by the authors is the ImpulseDE2 method (Fischer et al NAR 2018), which models temporal gene expression as "impulses". The package allows tests within condition as well as contrasting which genes follow different impulse functions between conditions. The lineage assignment can be seen as "condition" in this case. The authors of ImpulseDE2 have also released a single-cell specific version including ZINB called "LineagePule", but it is lacking in documentation.

It is true that the functionality in other methods only cover the screening test scenario in tradeSeq, but proper comparisons should still be made for this case.

Major issues

The different significance testing methods (tradeSeq, Monocle differentialGeneTest, GPFates OMGP likelihood ratio, ImpulseDE2) should be compared with the same input, be it from Slingshot or other, or directly from the simulation ground truths.

The default parameters for the spline regression based tests in Monocle use 3 knots, while tradeSeq uses 10. Repeating the analysis in Supp Fig S1 going all the way down to 3 knots would be interesting. And comparing results between Monocle and tradeSeq using the same number of knots.

Minor issues

It is not clear if the joint estimation of μ and ϕ or the shrinkage of the spline weights are features particular to tradeSeq or part of mgcv. They do not seem to immediately appear in the tradeSeq code.

It would be interesting to see the test applied to real time-course data as a proof of concept. It is not clear if any appropriate data exists, but demonstrating the method without the caveat that it depends on trajectory inference method would be of value.

Clustering of genes by using values predicted from the splines is also implemented in Monocle. Though it has not been demonstrated on bifurcating data. The RSEC based clustering and the K-medoids based clustering used in Monocle could be compared.

While not very important, it would be valuable to include a comparison of the runtimes of the different methods. It will let potential users calibrate their expectations when trying the methods themselves.

Reviewer #3:

Remarks to the Author:

Single-cell sequencing technologies enable the reconstruction of developmental trajectories, but the methods to identify differential expression patterns between lineages are still lacking. This paper developed tradeSeq to address this question. The method is built on gene-wise negative binomial generalized additive model (NB-GAM). Statistical testing on different trajectories should be interesting to most developmental biologists. Although they show the method outperforms

conventional methods in simulation studies, the validation in real data analysis is not very convincing. Major and minor revision comments are listed below.

Major ones:

I. Case study/ bone marrow

(1) For figure 3a & 3b, it is surprising not to see a clear separation between neutrophil and monocyte lineages by these dimension reduction methods. These two lineages are very different and can be clear separated by lineage analysis (such as Monocle: Nat Methods. 2017 Oct;14(10):979-982) or by PCA based on our research.

(2) In the section of Discovering cell type markers, they mentioned "Since the identification of a larger set of DE genes does not necessarily imply more relevant biology..." However, GSEA p-value is correlated with the number of input genes. The much lower p-value for tradeSeq analysis than BEAM is also associated with much more DEGs identified from tradeSeq. Besides, for tradeSeq, NES = 1.59 is not a high score. It may be difficult to see an obvious enrichment pattern (based on our results, visible enrichment is usually seen when NES ≥ 2). The authors should provide the GSEA plot for visualization. Only based on the result of 1 gene set, it is not sufficient to prove tradeSeq is superior to BEAM.

(3) It is also surprising that none of the methods recover the evidences for neutrophils. The authors should provide a reason for this. GO terms include a lot of items involved in neutrophils/ granulocytes/ monocytes, which are worth to try.

(4) In the section of Discovering progenitor population markers, this can be simply done by identify the makers genes based on clustering results and t-sne visualization, which is more straightforward.

2. Mouse olfactory epithelium dataset

(1) For within-lineage DE, Monocle has the similar function to identify the genes significantly associated with pseudotime. So it is not a novel function, either.

(2) For between-lineage DE, patternTest may be the most interesting function in tradeSeq. But validation, with just some gene set enrichment results, is not sufficient to show how robust their method is.

3. What is the computational complexity of the proposed method? Computing time and memory use should be discussed for both simulation and real studies.

4. For NB-GAM model, does the algorithm always converge? What about the ZINB model that was also discussed in the manuscript.

5. If the data are from multiple individuals with non-ignorable batch effects, how would the method handle it? How would individual information be incorporated into the model?

Minor ones:

1. The data visualization is weak throughout all the paper. For example, all the gene set enrichment results are in supplementary tables, which is not convenient and make paper difficult to read. Top or selected gene sets could be shown in main figures.

2. In general, the authors should focus on the patternTest function, which harbors the highest novelty in this paper, and provide sufficient evidences to demonstrate how robust it is.

Rebuttal: Trajectory-based differential expression analysis

September 1, 2019

1 Reviewer 1

The manuscript “Trajectory-based differential expression analysis” describes a statistical method and package for testing gene expression differences along a pseudotime trajectory. Trajectory reconstruction is a popular question for single-cell data and novel methods are constantly being developed, yet there is a need for a more general method of assessing significance and prioritizing genes. The approach described here is flexible and agnostic to any particular reconstruction method. Overall the manuscript is well-written with nice figures. The statistical framework used here is appropriate and shows good performance. The accompanying software package runs and would be useful to a broad audience. The results and conclusions presented are valid and well supported.

1.1 Code comments

1. I ran the exact same code in the vignette and obtained somewhat different results—in my case I only have a single lineage, which produced different plots and results in some tests not being available. I did use the same seed. Other than this difference, the code/package seems to work very well.

We appreciate the Reviewer’s efforts to evaluate our method and the reproducibility of our analyses. The fact that the results are not exactly identical is likely due to the updated random number generator in R version 3.6.0 and above, as compared to R version 3.5.0. This should be fixed, since we now specify the specific version of the random number generator in the preamble of the `tradeSeq` vignette.

1.2 Writing comments

1. There are a lot of subsections, which made it a bit difficult to follow completely through (this could just be the sizing and bold of every section since it was hard to tell where I was sometimes). More upfront motivation for why the two case-studies are analyzed differently might help with this – they do have completely different subsection titles.

While we hope that the flow will be clearer with the final lay-out and formatting of the manuscript, we have also tried to better motivate the distinct analyses of the two case studies by adding an overview paragraph for each. The relevant added text to the manuscript is shown below.

- Mouse bone marrow dataset, lines 580-585: *“In this case study, we apply `tradeSeq` with 6 knots, as found to be optimal by the AIC (Supplementary Figure S16). We first identify marker genes for the progenitor and differentiated cell types in the ‘Discovering cell type markers’ paragraph. Next, we assess which genes behave differently along the two lineages in the ‘Discovering progenitor population markers’ paragraph. Finally, we demonstrate how one can group genes in clusters that share similar expression patterns in the ‘Gene expression families’ paragraph.”*

- Mouse olfactory epithelium dataset, lines 664-666: “*In this case study, we first consider differential expression within each lineage in the ‘Within-lineage DE’ paragraph, after which we assess differences between the three developmental lineages in the ‘Between-lineage DE’ paragraph.*”
2. **The Introduction is well-written.** As a suggestion, the authors might consider a few sentences on the connection to somewhat similar underlying ideas for bulk RNA-seq time-series analysis methods. For example: <https://www.ncbi.nlm.nih.gov/pubmed/26046293>, <https://www.ncbi.nlm.nih.gov/pmc/articles/PMC4155246/>, <https://bmcbioinformatics.biomedcentral.com/articles/10.1186/s12859-018-2405-x>. Also, the single-cell method TSCAN uses a GAM to test for DE along the estimated trajectory, <https://academic.oup.com/nar/article/44/13/e117/2457590>. Obviously tradeSeq is much more extensive, but this paper should be cited.

We thank the Reviewer for the suggestion and have now included a section on the analogy with bulk RNA-seq time-series data analysis on lines 65-72:

“A number of methods have been developed for the analysis of bulk RNA-seq time-series data, which can exploit the continuous resolution of samples assayed at different times [Nueda et al., 2014, Sanavia et al., 2015, Bacher et al., 2018]. Nueda et al. [2014] requires multiple observations for each time-point and estimates the mean expression for each time-point. However, in scRNA-seq, cells are never at the exact same pseudotime value. Other approaches assume a piecewise linear [Bacher et al., 2018] or polynomial [Sanavia et al., 2015] relationship of mean expression with time, which provides insufficient flexibility to model the complex relationship between gene expression and pseudotime observed in scRNA-seq datasets. Often, these methods are also restricted to the estimation of only one or two functions for each gene.”

We now also discuss the original implementation of Monocle and TSCAN for within-lineage DE in the Introduction, on lines 73-77:

“A few methods have been published with the aim of improving trajectory-based differential expression analysis by modeling gene expression as a smooth function of pseudotime along lineages. Monocle [Trapnell et al., 2014] tests whether gene expression is associated with pseudotime by fitting additive models of gene expression as a function of pseudotime. However, the method can only handle a single lineage. A similar approach has been adopted by TSCAN [Ji and Ji, 2016].”

3. **The end of the Introduction has a sentence saying “one can build upon our stageR package”.** This should be in the Discussion or should be implemented within this new package. Did the authors mean, “we have built upon”?

The original sentence could indeed be misinterpreted. The stageR package is flexible and works by default with output from any trajectory-based differential expression method, as it only requires as input the p -values for the DE tests. Therefore, we have not included the stageR functionality in the tradeSeq package, but rather use it downstream of tradeSeq. We have rephrased the sentence on lines 113-116, which now states:

“If multiple hypotheses are assessed for each gene, one can use our stageR package [Van den Berge et al., 2017] to conduct an omnibus test (e.g., there are no differences in expression profiles across multiple lineages) prior to post hoc tests that identify the relevant specific differences (e.g., all pairwise comparisons between lineages).”

4. **Starting with the section “Within-lineage comparisons”** which lists the different tests, it would be incredibly useful to have an early figure with an example/cartoon of each of these cases highlighting the test points on the trajectory. I see the table, which is helpful but took me a bit longer to fully grasp. If space is limited in the main text, then this would be helpful in the vignette.

We thank the Reviewer for this comment. We have added an additional figure (Figure 1), which is a simpler version of the Table from the now Figure 2 in the manuscript. Figure 1 is also shown below.

Figure 1: *Current tests in the tradeSeq package.* Each column corresponds to a test. Tests are broken down into two categories, depending on whether they are a within-lineage comparison, hence testing properties of the orange curve, or a between-lineage comparison, contrasting the blue and orange curves. For each test, we have two toy examples of gene expression patterns. The top one would be differentially expressed according to the test while the bottom one would not.

5. In the Methods section on earlyDETest, it says “driving the differentiation”. I suggest a different phrasing here as that phrasing also connotes the specific biological process of differentiation and may be confusing to future readers.

We thank the Reviewer for pointing out the ambiguous explanation of the earlyDETest. We have rephrased this in the revised version of the manuscript on lines 245-246, which now states:

“The earlyDETest aims to identify genes that are differentiating around a branching of the trajectory.”

6. Page 10 the second paragraph on “Mouse bone marrow dataset” is confusing. It says, “We will therefore first apply ICA as dimensionality reduction method (Figure 3a) in the ‘Discovering cell type markers’ paragraph.” It would make more sense if it instead stated that the authors “first tried ICA” and explain the problems encountered as they have done, and then only use UMAP further.

We thank the Reviewer for this thoughtful comment and have now adapted the paragraph in the revised manuscript on lines 567-579. The revised text now reads:

“In order to compare our approach with BEAM, we are restricted to the dimensionality reduction procedures implemented in Monocle 2. We therefore first used ICA as dimensionality reduction method (Figure 5a) in the ‘Discovering cell type markers’ paragraph, but observed that this approach does not fully preserve the underlying biology. Indeed, a 2D visualization of the ICA dimensionality reduction shows that there is a seemingly large gap between the multipotent progenitors and the remaining cell types, and a number of erythrocytes and granulocyte-macrophage progenitors (GMP) are misclassified as multipotent progenitors. In addition, megakaryocytes, which are thrombocyte progenitors and as such should not belong to any of the two lineages, seem to be split between the erythrocyte and leukocyte lineages. However, when applying UMAP dimensionality reduction (Figure 5b), these issues are resolved and it seems to preserve the underlying biology better than ICA. In subsequent sections, we will therefore demonstrate the powerful interpretation of a tradeSeq.slingshot analysis based on UMAP dimensionality reduction. This additionally illustrates the flexibility of tradeSeq (and slingshot) to be applied downstream of any dimensionality reduction method.”

7. The package dyngen is referenced by name but no explanation is given, even just saying it is a simulation of dynamic trajectories would be helpful to future readers. I suggest the

authors double-check all package/methodology names are given a somewhat descriptive comment when introduced.

This has now been addressed in the revised version of the manuscript, where the `dyngen` package is explained, and we refer interested readers to the original `dyngen` paper.

On lines 351-356 and in the caption for Table 1, the revised manuscript now reads:

” The simulation study evaluates methods that (differentially) associate gene expression with pseudotime for three different trajectory topologies, i.e., a cyclic, a bifurcating, and a multifurcating trajectory. As independent evaluation, we use the extensive trajectory simulation framework `dynverse` that previously served for benchmarking trajectory inference methods in Saelens et al. [2019]. Interested readers should refer to the original publication for details on the data simulation procedure. Dataset characteristics are listed in Table 1. ”

Table 1: Overview of simulated datasets. Each dataset is simulated using one of the frameworks from the `dynverse` toolbox (`dyngen` or `dyntoy`), which are designed to simulate scRNA-seq data according to trajectory topologies. Every dataset can be characterized by the topology of the trajectory, as well as the number of cells and genes. Low-dimensional representations of representative datasets can be found in Figure 3. Note that the cyclic datasets have some variation in the numbers of genes and cells and in the amount of differential expression, which is inherent to the `dyngen` simulation framework.

	Cyclic dataset	Bifurcating dataset	Multifurcating dataset
Simulation framework	<code>dyngen</code>	<code>dyntoy</code>	<code>dyntoy</code>
Number of cells	505 – 508	500	750
Number of genes	312 – 444	5,000	5,000
% of DE genes	42 – 47%	20%	20%
Number of lineages	1	2	3
Topology	Cyclic	Bifurcating	Multifurcating
Number of datasets	10	10	1

1.3 Methods comments

1. **Why are different methods for enrichment used in the manuscript? Fgsea package and then the MSigDB database.**

For the mouse olfactory epithelium case study, we used the MSigDB database for exploratory gene set enrichment analysis, as in the original manuscript by Fletcher et al. [2017]. However, for the mouse bone marrow case study, we knew upfront which cell types (gene sets) should be found, and therefore selected carefully curated gene sets for gene set enrichment analysis. Unlike MSigDB, `fgsea` allows the use of custom gene sets, such as the gene sets from de Graaf et al. [2016].

2. **Why is the stage-wise testing only implemented for one case study dataset? Is it fair to compare to other methods only using the Benjamini Hochberg FDR control? Since the manuscript explicitly states that the p-values are only for ranking, it’s actually unclear the motivation for this extra step.**

Stage-wise testing is only relevant if multiple comparisons are of interest. The first case study consists of only one comparison (two lineages), hence stage-wise testing is not called for. In fact, stage-wise testing would simply reduce to conventional testing. For the second case study, there are three lineages and therefore three possible pairwise comparisons, hence stage-wise testing can be useful. Note that, whenever we compare methods, we make sure to adopt a similar testing strategy, e.g., in the comparison between the `diffEndTest` of `tradeSeq` and `edgeR` for the olfactory epithelium dataset, we only use a global test for both methods.

In the manuscript, we indeed write that p -values are mainly useful for ranking interesting genes. The

Benjamini-Hochberg FDR-controlling procedure preserves the unadjusted- p -value-based ranking of the genes, and we only use it to obtain a cut-off for particularly interesting genes, even though in practice the chosen cut-off might not coincide with the true, unknown FDR level.

3. **How should future readers/users think of knots? As just markers of relative timing along the trajectory? For example, Figure 4 says the DE test was done between knots 2 and 4. How to interpret this choice?**

In `tradeSeq`, the knots are positioned according to the quantiles of the pseudotime values. For example, if smoothers are fit with 3 knots, then there will be a knot at the minimum, median, and maximum pseudotime values. If pseudotime values are accurately estimated, the knots may indeed be interpreted as markers of relative timing along the trajectory. However, it is important to realize that pseudotime might not necessarily linearly correlate with true chronological time. Indeed, a large distance in gene expression space can occur in a short span of true chronological time, and vice versa. We have now incorporated a discussion of this issue in the 'Methods' section on lines 183-187 in the revised manuscript: *“The knots are by default positioned according to the quantiles of the pseudotime values. For example, if a smoother is fit with 3 knots, then there will be a knot at the minimum, median, and maximum pseudotime values. The knots may be interpreted as relative markers of progress along the trajectory. However, it is important to realize that this might not necessarily linearly correlate with true chronological time.”*

We have also implemented additional functionality in `tradeSeq` that aids users in selecting an appropriate number of knots, see Section 'Choosing an appropriate number of knots' on line 177 of the revised manuscript. Furthermore, `tradeSeq` has built-in plotting functions that allow the user to visualize the knots, as demonstrated in our vignette.

In order to motivate our choice of knot points between which to perform the `earlyDETest` in the olfactory epithelium case study dataset, we choose a region of the trajectory that contains the branching event we are interested in. Supplementary Figure S22 (shown below) now shows the trajectory where the locations of the knot points have been plotted on each lineage. Note that in the revision we have now re-analyzed the dataset with 6 knots as determined using the AIC. The branching event now lies between knot points 1 and 3, and therefore we choose to adopt the `earlyDETest` between these knot points. In order to better argument this selection, we have now incorporated this Figure in the Supplementary Figures, and refer to it in the analysis of the olfactory epithelium case study on lines 716-717 in the revised manuscript:

“We can also identify genes that drive the branching based on the `earlyDETest` applied around the first branching point, i.e. between knots 1 and 3 (see Supplementary Figure S22 and Figure 6d).”

Supplementary Figure 22: *Olfactory epithelium trajectory with knots*. Three-dimensional PCA plot of the scRNA-seq data from Fletcher et al. [2017], where cells are colored according to their cluster membership as defined in the original paper (see Methods). The simultaneous principal curves for the lineages inferred by slingshot are displayed. The numbers on each lineage specify the knot points used to fit the ZINB-GAM in tradeSeq. The first branching event occurs between knots 1 and 3.

4. Are the datasets pre-filtered at all or was a GAM fit for every gene?

The datasets in the simulation study are not filtered. Datasets in both case studies have been filtered to contain genes with reasonable expression levels. We have now specified this in the 'Methods' section of the specific case studies.

- Bone marrow dataset, on lines 379-380: “We use the same dataset from the Monocle 3 vignette, which has been prefiltered to contain genes with relatively high expression. Upon filtering, the dataset consists of 3004 genes and 2660 cells.”
- Olfactory epithelium dataset, on lines 400-402: “Prior to the analysis, the dataset is filtered for genes with reasonably high expression, and we consider 14,261 genes and 616 cells for downstream analysis.”

1.4 Figure comments

1. Figure 1 says “knots = c(1, 2).” but it’s unclear what that actually means.

We have now addressed this in the revised version of the manuscript, where the relevant part of the caption for the current Figure 2 (which used to be Figure 1 in the original version of the manuscript) now states:

“In the table, we assume that the *earlyDETest* is used to assess differences in expression patterns early

in the lineage, e.g., with option `knots = c(1, 2)`, meaning that we test for differential patterns between the first and second dashed grey lines from panel b.”

- 2. Figure 3. How is the clustering done here? In the six clusters, I assume each gene has a green and an orange line, but how are they so consistent within a cluster? Are the data-points used to cluster pulled from both lineages?**

Yes, first a GAM is fitted for each gene. Subsequently, the estimated profiles for both lineages are extracted for each gene and the genes are then clustered based on their estimated expression profiles in both lineages. Hence, genes that are clustered together are assumed to have a similar expression profiles in all lineages of the trajectory. We refer the Reviewer to the ‘Methods’ section on the clustering, where on lines 283-284 of the revised manuscript, the following is stated:

“The NB-GAM can also be used to cluster genes according to their expression patterns, as shown in Figure 1. Specifically, for each gene, we extract a number of fitted values for each lineage (100 by default).”

2 Reviewer 2

The authors describe a strategy called tradeSeq to analyze per-lineage gene expression patterns in scRNA-seq trajectory data. In particular they have identified a set of hypothesis tests to query for particular smooth within-lineage trends and cross-lineage comparisons. To account for multiple testing the authors make clever use of the theory of stage-wise testing.

To model the data weighted spline regression with negative binomial likelihood is used, this is implemented as a wrapper around the mgcv package. The strategy is similar to the strategy implemented in the Monocle package.

While the choice of model likelihood is motivated by scRNA-seq data, there is nothing in particular about the model used which relates to the predictors being “pseudotime” and lineage membership.

In this study the authors also perform a comparison between pseudotime and lineage assignment methods identifying Slingshot as performing well.

Finally, case studies of using the different hypothesis tests in tradeSeq to answer questions about transient and lineage-specific expression patterns are demonstrated on published bone marrow and olfactory epithelium data.

The interpretable significance tests are solving a severe problem in the community. However, the underlying model is not correctly compared to the alternative published methods, and the authors claim other methods can only be applied in particular scenarios of upstream analysis, which is only true on a highly superficial level.

The method presented by the authors requires as input a gene expression matrix, per-cell pseudotime, and per-cell lineage membership. The same can be used with other methods.

In Monocle’s BEAM test, the significance test is performed by comparing a spline GLM which includes an interaction between the spline parameters and the lineage membership with one without interaction. It is implemented as a wrapper around VGAM similar to how tradeSeq wraps mgcv. In tradeSeq regression weights are used instead of making the interaction part of the model, but the effect will be the same. The same underlying model can be fitted in Monocle using the differentialGeneTest method (which can also use other covariates such as batch).

The authors claim edgeR can only be used for discrete tests, but edgeR supports arbitrary design matrices, including ones made with splines, equivalent to the tradeSeq model or Monocle’s BEAM models.

The bifurcation test in GPFates performs a likelihood ratio calculation of two OMGP models: one with pre-fitted cell lineage memberships, and one with equal membership to each trajectory. This test could also be performed with the inputs specified for tradeSeq.

Not mentioned by the authors is the ImpulseDE2 method (Fischer et al NAR 2018), which models temporal gene expression as “impulses”. The package allows tests within condition

as well as contrasting which genes follow different impulse functions between conditions. The lineage assignment can be seen as “condition” in this case. The authors of ImpulseDE2 have also released a single-cell specific version including ZINB called “LineagePule”, but it is lacking in documentation.

It is true that the functionality in other methods only cover the screening test scenario in tradeSeq, but proper comparisons should still be made for this case.

We thank the Reviewer for their constructive comments. We agree with the Reviewer that the mean model formulation of tradeSeq is similar to that of Monocle. However, we find improved results with tradeSeq as compared to Monocle, both in the simulation study as well as in the case study. This could be an effect of improved estimation of the dispersion parameter of the negative binomial distribution. Indeed, Monocle adopts a method-of-moments estimator for the dispersion parameter, which is quick but statistically inefficient.

As suggested by the Reviewer, it is possible to use a basis function expansion as design matrix in traditional DE methods like edgeR. However, we have decided to not evaluate this approach, since the edgeR model does not allow for shrinkage of the regression coefficients. Hence, edgeR would be prone to overfitting since no regularization is performed to fit the smoothers. We have now added this to the manuscript when we describe the edgeR model, on lines 344-349:

“ Note that, while edgeR was originally developed for group-based differential expression, it would be possible to incorporate the basis functions of the smoothers as continuous covariates in the model. However, no regularization would be performed on the estimation of the smoother regression coefficients, hence the model would be prone to overfitting. A similar approach was evaluated in Fischer et al. [2018], where DESeq2 [Love et al., 2014] was used to fit splines by incorporating natural cubic basis functions in the model. ”

The inference procedure applied by GPfates is indeed somewhat different from the procedure applied by tradeSeq. With tradeSeq, we are able to build on recent theoretical developments for inference on smoother coefficients [Wood, 2012] which, to our knowledge, are not available for Gaussian process regression. Because of this, GPfates must rely on a computer-intensive permutation scheme, which hampers it in terms of scalability, as confirmed by our novel computation time benchmark in the revised version of the manuscript. The difference in inference methodology is therefore beneficial for tradeSeq.

We thank the Reviewer for pointing us to the ImpulseDE2 method and we have now included it in our evaluations based on the simulated datasets. The method, however, does not perform as well as most other methods. In addition, it requires a high computation time.

2.1 Major issues

1. **The different significance testing methods (tradeSeq, Monocle differentialGeneTest, GP-Fates OMGP likelihood ratio, ImpulseDE2) should be compared with the same input, be it from Slingshot or other, or directly from the simulation ground truths.**

In the simulation study, we have now, as much as possible, compared methods on equal grounds.

First, in the bifurcating trajectory simulation, we have now additionally compared all methods based on the simulation ground truth, as suggested by the Reviewer. Qualitatively, the results are consistent with our previous comparison, with tradeSeq outperforming the other methods. These results naturally extend to the multifurcating trajectory setting. The relevant paragraph, on lines 507-515 of the revised manuscript, states:

“ In order to avoid the comparison of DE methods being obscured by differences in the upstream dimensionality reduction and trajectory inference methods, we compared tradeSeq, BEAM, and ImpulseDE2 on the simulation ground truth. We fit the tradeSeq NB-GAM once with 3 knots, for comparability with the BEAM approach that also uses 3 knots, and once with 4 knots, which was found to be optimal according to the AIC (Supplementary Figure S7). The tradeSeq patternTest is unaffected by the change in the number of knots and outperforms all other methods for differential expression analysis (Supplementary Figure S12). The performance of the tradeSeq diffEndTest is somewhat sensitive to the number of knots, but still better than that of ImpulseDE2 and BEAM. Generally, ImpulseDE2 performs better than the BEAM approach. ”

Second, in the cyclic trajectory simulation, we have compared `tradeSeq_slingshot` with Monocle 3 once using the default dimensionality reduction technique for each method (PCA for `slingshot` and UMAP for Monocle 3), and once using UMAP dimensionality reduction for both methods. Note that the Moran's I test that is implemented in Monocle 3 does not use information on the trajectory, only on the reduced dimensional representation of the cells, which obviates a comparison based on simulation ground truth. The relevant paragraph, on lines 470-474 of the revised manuscript, states:

“We also compared both methods using the same dimensionality reduction input, by having `slingshot` infer trajectories in the UMAP space that is used by Monocle 3. The performance of `tradeSeq` was generally similar for both dimensionality reduction methods, except for 2 out of 10 datasets (Supplementary Figure S6). In all datasets, `tradeSeq` had better performance than Monocle 3.”

- 2. The default parameters for the spline regression based tests in Monocle use 3 knots, while `tradeSeq` uses 10. Repeating the analysis in Supp Fig S1 going all the way down to 3 knots would be interesting. And comparing results between Monocle and `tradeSeq` using the same number of knots.**

We thank the Reviewer for their thoughtful comment, which has led us to further extend `tradeSeq`. The analysis in Supplementary Figure S1 has now been repeated when varying the number of knots k down to 3 knots, as shown below. Interestingly, whereas the distribution of explained deviance remains similar for most values of k , it is lower when using only 3 knots, which is the minimum number of knots one can use.

This has led us to develop diagnostic plots to help future users of `tradeSeq` to decide on an optimal number of knots k for the dataset at hand. This extra functionality is described in a new Methods section, ‘Choosing an appropriate number of knots’ on line 177 of the revised manuscript, which we copied below. We have subsequently proceeded to use this additional functionality to select an appropriate number of knots in the analysis of all simulated and real datasets in the manuscript. Following this evaluation, we have now changed the default number of knots for `tradeSeq` to 6, instead of 10.

Choosing an appropriate number of knots

“ Ideally, the number of knots K should be selected to reach an optimal bias-variance trade-off for the smoother, where one explains as much variability in the expression data as possible with only a few regression coefficients (see Supplementary Figure S1). In practice, the optimal number of knots K may be selected by evaluating the Akaike Information Criterion (AIC) using the `evaluateK` function implemented in `tradeSeq`. We have deliberately chosen for the AIC as evaluation criterion, since the Bayesian Information Criterion (BIC) seemed to favour too complex models (i.e., suggesting an excessively high number of knots). The knots are by default positioned according to the quantiles of the pseudotime values. For example, if a smoother is fit with 3 knots, then there will be a knot at the minimum, median, and maximum pseudotime values. The knots may be interpreted as relative markers of progress along the trajectory. However, it is important to realize that this might not necessarily linearly correlate with true chronological time.”

Supplementary Figure 1: *Mouse bone marrow dataset: The NB-GAM is robust to the number of knots k .* Gaussian kernel density plot of the percentage of deviance explained by the NB-GAM applied to each of the genes in the dataset from Paul et al. [2015], with number of knots k ranging from 3 to 14. The distributions are nearly identical for the different numbers of knots, except for 3 knots, suggesting we might want to select more than 3 knots for this dataset.

2.2 Minor issues

1. **It is not clear if the joint estimation of mu and phi or the shrinkage of the spline weights are features particular to tradeSeq or part of mgcv. They do not seem to immediately appear in the tradeSeq code.**

The joint estimation feature for the mean and dispersion parameters of the negative binomial model is a recent development of the `mgcv` package and it is not a novelty implemented by `tradeSeq`. We have now adapted the text in 'Methods' on lines 164-166 of the revised manuscript to clearly specify this: "The NB-GAM is fitted gene by gene using the `fitGAM` function from the `tradeSeq` package, which relies on the `mgcv` package in R. We build upon recent developments in `mgcv` that allow the joint estimation of the NB regression parameters in μ_{gi} and of the dispersion parameter ϕ_g [Wood et al., 2016]."

2. **It would be interesting to see the test applied to real time-course data as a proof of concept. It is not clear if any appropriate data exists, but demonstrating the method without the caveat that it depends on trajectory inference method would be of value.**

We have now also demonstrated `tradeSeq`'s applicability beyond scRNA-seq data, but applying it to the bulk RNA-seq time-course study of Kiselev et al. [2015]. We have compared the `tradeSeq` results with the results in the original manuscript and we now briefly describe this as one of the case studies in the revised manuscript, on line 551. For reference, we include the text from the revised manuscript below:

Bulk RNA-seq time-course study

“ While in this manuscript we focus on DE analysis downstream of TI, the applicability of `tradeSeq` extends beyond this setting. We demonstrate this by using `tradeSeq` on a bulk RNA-seq time-course study from Kiselev et al. [2015], where we compare gene expression between wild type and `PIK3CA H1047R` cell lines upon stimulation of epidermal growth factor (EGF). Gene expression was measured for three replicates in each condition over over six time-points, spanning from 0 to 300 minutes post EGF stimulation. The original analysis in the manuscript assessed DE between the cell lines for each time-point separately using `DESeq2`, and found 7,486 DE genes at a 1% nominal FDR level. We perform an analogous analysis using `tradeSeq`, by modeling gene expression measures as smooth functions of time and looking for differences in expression patterns with `patternTest`. This yields 7,184 DE genes at a 1% nominal FDR level. Around 89% of these genes overlap with the original DE list of 7,486 genes, demonstrating that the utility of `tradeSeq` goes beyond scRNA-seq applications.”

3. **Clustering of genes by using values predicted from the splines is also implemented in Monocle. Though it has not been demonstrated on bifurcating data. The RSEC based clustering and the K-medoids based clustering used in Monocle could be compared.**

We thank the Reviewer for their constructive comment. However, we note that a comparison between RSEC and K-medoids clustering (i.e., partitioning around medoids or PAM) might be cumbersome, for two reasons. First, the Monocle clustering will rely on the smoothers estimated by Monocle, while the `tradeSeq` clustering will rely on the `tradeSeq` models. Since the fitted values of these smoothers will be different, their clustering will be different, even if the same clustering algorithm is used. A comparison between RSEC and PAM will thus be confounded by the fitted models, since it will be impossible to know whether a ‘better clustering’ is a consequence of the difference in models or in clustering algorithms. Additionally, RSEC encompasses the PAM clustering algorithm, and a comparison of consensus clustering vs. clustering by individual algorithms is extensively discussed in the RSEC manuscript [Risso et al., 2018a]. Indeed, RSEC provides a range of different clustering algorithms (e.g., PAM and k -means) and allows the user to derive a consensus clustering across all selected clustering algorithms and a range of required tuning parameter values for each algorithm. Since the clustering function implemented within `tradeSeq` wraps around the RSEC function of `clusterExperiment`, it actually allows users to adopt the clustering algorithm of their choice. This means that, if the user selects the PAM algorithm for clustering with `tradeSeq`, the clustering method reduces to PAM clustering, just like in Monocle. However, `tradeSeq` also allows for more advanced options, such as deriving a consensus clustering across several different clustering algorithms and their corresponding tuning parameters. We have now more clearly specified this in the ‘Clustering gene expression patterns’ subsection of the ‘Methods’ section on lines 283-291, which is pasted below:

“The NB-GAM can also be used to cluster genes according to their expression patterns, as shown in Figure 2. Specifically, for each gene, we extract a number of fitted values for each lineage (100 by default). We can then use resampling-based sequential ensemble clustering (RSEC), as implemented in `clusterExperiment` [Risso et al., 2018a], to perform the clustering based on the first ten principal components of the standardized fitted values matrix (i.e., the fitted values are standardized to have zero mean and unit variance across cells for each gene). Importantly, we allow for any clustering algorithm that is built-in into `clusterExperiment` or chosen by the user to perform the clustering. This clustering approach is implemented in the `tradeSeq` package (`clusterExpressionPatterns` function) for downstream analysis facilitating the interpretation of DE genes.”

4. **While not very important, it would be valuable to include a comparison of the runtimes of the different methods. It will let potential users calibrate their expectations when trying the methods themselves.**

We thank the Reviewer for this constructive comment. We have added a subsection in the ‘Results’ section on line 528, called ‘Computation time and memory benchmark’, which is pasted below:

” To assess time and memory requirements, scRNA-seq datasets with a bifurcating trajectory were simulated using the same simulation framework as in the simulation study. Three datasets with 100, 1,000, and 10,000 cells were simulated (small, medium, and large datasets), each consisting of 5,000 genes. For BEAM and GPfates, only the fitting and DE testing part was assessed for each method, not the trajectory inference part. All methods were ran with default options. For tradeSeq, the `fitGAM` function was assessed with 4 knots, as determined in the simulation study. The different tests implemented in tradeSeq were benchmarked separately (Supplementary Figure S15). Their running times are very small (always below 30 seconds) as compared to the `fitGAM` function, and do not increase for datasets with increasing numbers of cells.

To benchmark time requirements, the `microbenchmark` package was used, and each method was run 10, 2, and 2 times on respectively the small, medium, and large datasets. Variations in running times were very small (always under the minute), especially in comparison to between methods differences. Jobs that reached the 4-hour mark without finishing were killed. This is the case for `ImpulseDE2` on the datasets of 10^3 and 10^4 cells, and for `GPfates` on the largest dataset. Memory benchmark was assessed using the `Rprof` function. Maximum memory usage was recorded.

Results can be seen in Figure 3. `ImpulseDE2` is by far the slowest, taking over 3.5 hours to finish on a small dataset of 100 cells. `GPfates` runs in about 30s on the small dataset but scales poorly. `BEAM` and `edgeR` are quite fast and scale very well, even to large datasets. `tradeSeq` is slower but can still handle the larger datasets reasonably well. In term of memory requirements, all methods scale well to 10,000 cells. It should also be noted that `tradeSeq`, `ImpulseDE2`, and `BEAM` can utilize multiple cores but were benchmarked using only one core.”

Figure 3: Benchmark of computation time and memory usage. Datasets for 100, 1000 and 10000 cells are simulated and each method is evaluated respectively 10, 2 and 2 times on each dataset to assess memory and computation time usage. The average across iterations is plotted for each method. Methods that went over a 4-hour mark were stopped and deemed taking too long. Source Data are provided as a Source Data File.

Supplementary Figure S15: *Computational time benchmark for the various tests implemented in tradeSeq.* Datasets of increasing size (in terms of number of cells) were simulated, each consisting of 5000 genes. The `fitGAM` function of `tradeSeq` was ran with 4 knots. The computational time required to run the tests for all genes is benchmarked using the `microbenchmark` package, with 10 iterations each. The `patternTest` and `earlyDETest` are slower than the `associationTest`, `diffEndTest`, and `startVsEndTest`, but all take under 30 seconds to run. The time requirement is constant with respect to the number of cells.

3 Reviewer 3

Single-cell sequencing technologies enable the reconstruction of developmental trajectories, but the methods to identify differential expression patterns between lineages are still lacking. This paper developed `tradeSeq` to address this question. The method is built on gene-wise negative binomial generalized additive model (NB-GAM). Statistical testing on different trajectories should be interesting to most developmental biologists. Although they show the method outperforms conventional methods in simulation studies, the validation in real data analysis is not very convincing. Major and minor revision comments are listed below.

The Reviewer raises several interesting issues that have been addressed in the revised manuscript. We would also like to emphasize that the aim of our manuscript is not to develop novel statistical testing procedures. Rather, we contribute a rigorous and robust statistical framework for trajectory-based differential expression that, unlike previous approaches, provides a clear and biologically meaningful interpretation of DE as it relates to the trajectory.

The Reviewer correctly points out that analogous versions of several statistical tests that are provided by `tradeSeq` are indeed available in other software packages. We provide a software package that combines previously developed tests and, additionally, implements novel testing procedures, leading to clear interpretability of the results in the context of differential expression along or between lineages of a trajectory.

The proposed testing procedures are deeply rooted in the generalized additive model (GAM) framework, which has allowed us to unify all previously developed tests. Indeed, the proposed GAMs allow for generalizations of group-based (i.e., discrete) differential expression, i.e., comparing groups of cells along the trajectory (e.g., `tradeSeq`'s `startVsEndTest`, `diffEndTest`), as well as more complex tests, such as assessing patterns of differential expression (e.g., `tradeSeq`'s `patternTest`). Notably, through its modularity, `tradeSeq` avoids ‘pipeline lock-in’, where some previously developed statistical tests were only available downstream of a specific trajectory inference method.

We use the simulation study as an extensive benchmark of `tradeSeq` vs. other available group-based or trajectory-based differential expression methods. This simulation study has been extended in the revised version of our manuscript. While the Reviewer raises benchmarking and methodological comments in the context of our case studies, we argue that some of these comments are addressed by our extensive simulation study of over twenty datasets spanning three different trajectory topology classes. Indeed, comparing methods on real datasets where limited to no ground truth is known for gene-level differential expression would make for an insufficient benchmark. Our case studies illustrate the flexibility of `tradeSeq` when applied to real datasets, rather than serve as a thorough benchmark of differential expression procedures.

3.1 Major ones

3.1.1 Case study/ bone marrow

1. **For figure 3a & 3b, it is surprising not to see a clear separation between neutrophil and monocyte lineages by these dimension reduction methods. These two lineages are very different and can be clear separated by lineage analysis (such as Monocle: Nat Methods. 2017 Oct;14(10):979-982) or by PCA based on our research.**

We thank the Reviewer for this thoughtful comment, although we note that other authors have also not found a clear separation. Please note that, aside from the Paul et al. [2015] dataset that we have used as a case study, the Monocle 2 paper mentioned by the Reviewer [Qiu et al., 2017] also describes a second dataset that investigates myelopoiesis [Olsson et al., 2016]. In this dataset, there is indeed a clear separation between granulocytes (which contain neutrophils) and monocytes, e.g., Figure 2 of the paper mentioned by the Reviewer [Qiu et al., 2017]. For the dataset that we have analyzed [Paul et al., 2015], the Monocle 2 tutorial and paper [Qiu et al., 2017] also do not recover a separation between the neutrophil and monocyte cell lineages. For example, this is clearly shown in Supplementary Figure 16, panel B of the Monocle 2 paper [Qiu et al., 2017], where many neutrophils and monocytes are embedded in the same lineage of the inferred trajectory. This is again confirmed by the vignette of Monocle 3. Hence, the separation of the two cell types seems to depend on the dataset, and it is not a consequence of the dimensionality reduction method used, since no separation is found using either ICA (Figure 5a in the revised manuscript), Monocle 2 (which uses DDRTRee dimensionality reduction), nor Monocle 3 (UMAP dimensionality reduction). Thus, it also does not affect our analyses with `tradeSeq`.

2. **In the section of Discovering cell type markers, they mentioned “Since the identification of a larger set of DE genes does not necessarily imply more relevant biology...” However, GSEA p-value is correlated with the number of input genes. The much lower p-value for `tradeSeq` analysis than BEAM is also associated with much more DEGs identified from `tradeSeq`. Besides, for `tradeSeq`, NES = 1.59 is not a high score. It may be difficult to see an obvious enrichment pattern (based on our results, visible enrichment is usually seen when NES >= 2). The authors should provide the GSEA plot for visualization. Only based on the result of 1 gene set, it is not sufficient to prove `tradeSeq` is superior to BEAM.**

The GSEA approach that we have used here only relies on the ranking of the genes to determine enrichment. Since every method investigates an equal number of genes (i.e., all genes in the dataset), there is no bias in the GSEA p -values due to different degrees of differential expression evidence between the methods. We have now updated the manuscript to include the enrichment plots for all methods (i.e., `tradeSeq`, BEAM, and edgeR). The relevant text, on lines 596-612 in the revised manuscript, and Figure

are provided below.

“ Since the identification of a larger set of DE genes does not necessarily imply more relevant biology, we select carefully constructed gene sets from de Graaf et al. [2016] to perform gene set enrichment analysis (GSEA) on blood cell types. As we are comparing erythrocytes with a mixture of leukocytes, we expect gene sets related to erythrocytes to be significant. Indeed, the erythrocyte gene set is the only one to be found significant by fgsea [Sergushichev, 2016] for the `tradeSeq` analysis (FDR adjusted p -value $< .001$, with normalized enrichment score of 1.49), while no significant gene sets are found for the BEAM analysis (as reference, in that case, the FDR adjusted p -value for the erythrocyte gene set is 0.58). In this case, `tradeSeq` is therefore better able to recover a meaningful biological signal (Supplementary Figure S18).

If one assumes that the cell type labels are known for all cells in the dataset, a cluster-based comparison is possible, where the different clusters correspond to the identified cell types. We use `edgeR` [McCarthy et al., 2012] to assess differential expression between erythrocytes and neutrophils, since this comparison is most analogous to `tradeSeq`'s `diffEndTest`. Only `edgeR` finds evidence for gene sets related to eosinophils and T-cells (FDR adjusted p -values of 0.042 and 0.049, respectively), however, the eosinophil cells were removed from this dataset prior to analysis (see Methods, subsection ‘Case studies: Mouse bone marrow dataset’). The GSEA results for `edgeR` also provide less evidence for erythrocytes (FDR adjusted p -value = 0.043, normalized enrichment score=1.21) as compared to the `tradeSeq` analysis (Supplementary Figure S18). ”

Supplementary Figure S18: Mouse bone marrow dataset: Gene set enrichment plots for the erythrocyte gene set from de Graaf et al. [2016], for three differential expression methods: `tradeSeq`, `BEAM`, and `edgeR`. Enrichment is determined for each method based on its respective ranking of the genes according to evidence for differential expression. Genes that are contained in the erythrocyte gene set are denoted with vertical lines at the bottom of each Figure, and the green curve represents the gene enrichment score along the gene rankings. `tradeSeq` has the highest enrichment score, as determined by the dashed red line, since genes that are related to erythrocytes predominantly have high rankings for differential expression, while the distribution of erythrocyte genes seems more uniform with, for example, the `BEAM` approach.

3. It is also surprising that none of the methods recover the evidences for neutrophils. The authors should provide a reason for this. GO terms include a lot of items involved in neutrophils/ granulocytes/ monocytes, which are worth to try.

We hypothesize that this is also due to possible confusion of the different datasets that have been used in the original Monocle paper. There seems to be a separation in the dataset from Olsson et al. [2016], while this is not obvious in the dataset from Paul et al. [2015], both according to our analysis and the analysis from Qiu et al. [2017].

4. In the section of Discovering progenitor population markers, this can be simply done by identify the makers genes based on clustering results and t-sne visualization, which is

more straightforward.

In our manuscript, we argue that cluster-based comparisons in dynamic biological systems are sub-optimal. In the subsection on ‘Discovering progenitor population markers’, we assess two hypotheses: (a) identifying marker genes that are differentially expressed between the progenitor population and the differentiated cell types; (b) identifying genes that are activated during development, but are not differentially expressed between the progenitor population and differentiated cell types. The hypothesis of (a) might indeed similarly be assessed by comparing clusters of the progenitor cell population vs. the differentiated cell types, which is also what we describe in the ‘Methods’ section of the `startVsEndTest` procedure on lines 219-222 in the revised manuscript:

“Note that contrasting the start and endpoints of a lineage is a special case of a more general capability of `tradeSeq` to compare the average expression between any two regions of a given lineage. As such, this test can be considered a generalization of cluster-based discrete DE within a lineage (e.g., [Risso et al., 2018b]).”

However, the second hypothesis (b) is far more difficult to assess using cluster-based comparisons. Indeed, it is unclear which clusters should be compared if one wants to assess gene expression differences during development, rather than at the start vs. end stage of a trajectory. Importantly, the activation may also happen within a single cluster, which would be missed with cluster-based comparisons, but not with the flexible GAM models that are proposed in our manuscript. Thus, while we agree that visualization of interesting genes by coloring cells in t-SNE dimensions according to their respective expression levels is a useful tool, we do not believe that the hypotheses that are considered in the ‘Discovering progenitor population markers’ section could be evaluated by looking at t-SNE plots of clustering results.

3.1.2 Mouse olfactory epithelium dataset

1. **For within-lineage DE, Monocle has the similar function to identify the genes significantly associated with pseudotime. So it is not a novel function, either**

While we acknowledge that an analogous version of `tradeSeq`’s `associationTest` is implemented in `Monocle`, we would like to stress several advantages of our approach: (i) The `Monocle` test only allows to assess association of gene expression levels and pseudotime across the entire trajectory, while we allow for lineage-specific assessment of association of expression and pseudotime; (ii) the `Monocle` test is implemented in a way that, currently, is only compatible downstream of `Monocle` trajectory inference, while `tradeSeq` is modular and can be applied downstream of any trajectory inference method; (iii) the `tradeSeq` `associationTest` is more powerful than the `Monocle` approach, as demonstrated in our simulation study.

2. **For between-lineage DE, `patternTest` may be the most interesting function in `tradeSeq`. But validation, with just some gene set enrichment results, is not sufficient to show how robust their method is.**

The purpose of the case studies is not to demonstrate the robustness of our methods. Instead, we refer the Reviewer to our extensive simulation study, where we have thoroughly benchmarked all methods. The case studies only illustrate the flexibility of our method. In particular, `tradeSeq` is the only trajectory-based differential expression analysis method that can be applied to the mouse olfactory epithelium dataset, since (i) this dataset is zero-inflated and (ii) the trajectory consists of more than two lineages. With `tradeSeq`, we are able to prioritize potentially interesting candidate genes for downstream biological validation, while none of the other methods would be capable of appropriately analyzing this dataset. While biological validation of our suggested results is very interesting from a biological perspective, this is not the main goal of our manuscript.

3. **What is the computational complexity of the proposed method? Computing time and memory use should be discussed for both simulation and real studies.**

We thank the Reviewer for this constructive comment. We have added a subsection in the 'Results' section on line 528, called 'Computation time and memory benchmark', which is pasted below:

“ To assess time and memory requirements, scRNA-seq datasets with a bifurcating trajectory were simulated using the same simulation framework as in the simulation study. Three datasets with 100, 1,000, and 10,000 cells were simulated (small, medium, and large datasets), each consisting of 5,000 genes. For BEAM or GPFates, only the fitting and DE testing part was assessed for each method, not the trajectory inference part. All methods were ran with default options. For tradeSeq, the `fitGAM` function was assessed with 4 knots, as determined in the simulation study. The different tests implemented in tradeSeq were benchmarked separately (Supplementary Figure S15). Their running times are very small (always below 30 seconds) as compared to the `fitGAM` function, and do not increase for datasets with increasing numbers of cells.

To benchmark time requirements, the `microbenchmark` package was used, and each method was run 10, 2, and 2 times on respectively the small, medium, and large datasets. Variations in running times were very small (always under the minute), especially in comparison to between methods differences. Jobs that reached the 4-hour mark without finishing were killed. This is the case for `ImpulseDE2` on the datasets of 10^3 and 10^4 cells, and for `GPFates` on the largest dataset. Memory benchmark was assessed using the `Rprof` function. Maximum memory usage was recorded.

Results can be seen in Figure 3. `ImpulseDE2` is by far the slowest, taking over 3.5 hours to finish on a small dataset of 100 cells. `GPFates` runs in about 30s on the small dataset but scales poorly. `BEAM` and `edgeR` are quite fast and scale very well, even to large datasets. `tradeSeq` is slower but can still handle the larger datasets reasonably well. In term of memory requirements, all methods scale well to 10,000 cells. It should also be noted that `tradeSeq`, `ImpulseDE2`, and `BEAM` can utilize multiple cores but were benchmarked using only one core.”

Figure 3: *Benchmark of computation time and memory usage.* Datasets for 100, 1000 and 10000 cells are simulated and each method is evaluated respectively 10, 2 and 2 times on each dataset to assess memory and computation time usage. The average across iterations is plotted for each method. Methods that went over a 4-hour mark were stopped and deemed taking too long. Source Data are provided as a Source Data File.

Supplementary Figure S15: *Computational time benchmark for the various tests implemented in tradeSeq.* Datasets of increasing size (in terms of number of cells) were simulated, each consisting of 5000 genes. The `fitGAM` function of `tradeSeq` was ran with 4 knots. The computational time required to run the tests for all genes is benchmarked using the `microbenchmark` package, with 10 iterations each. The `patternTest` and `earlyDETest` are slower than the `associationTest`, `diffEndTest`, and `startVsEndTest`, but all take under 30 seconds to run. The time requirement is constant with respect to the number of cells.

4. **For NB-GAM model, does the algorithm always converge? What about the ZINB model that was also discussed in the manuscript.**

The fitting procedure for the NB-GAM is not guaranteed to converge for every single gene in a dataset. Convergence very much depends on the filtering of the dataset. If strong filtering has been adopted and only genes with relatively high expression are retained, the procedure typically converges well. The NB-GAM is typically much harder to fit for genes with very low expression (e.g., zero expression in all but a few cells). The fitting also becomes more difficult with more lineages in the trajectory and when incorporating zero inflation weights. In our case studies, we have filtered out the genes with very low expression, improving the fitting of the models. For example, the fitting procedure converged for all genes in the mouse bone marrow dataset from Paul et al. [2015], while for the mouse olfactory epithelium dataset of Fletcher et al. [2017], where three lineages are present and zero inflation weights are adopted, the fitting failed for 0.8% (120 out of 14,261) of the genes. In the revised version of the manuscript, we have added this information to the introductory paragraph of the mouse olfactory epithelium case study in the 'Results' section at lines 660-661 in the revised manuscript, which states: “We were unable to fit a model for 0.8% of all 14,261 genes due to convergence issues of the ZINB-GAM.”

5. **If the data are from multiple individuals with non-ignorable batch effects, how would the method handle it? How would individual information be incorporated into the model?**

If data are from multiple individuals, `tradeSeq` allows to incorporate the individual effect as a fixed effect in the model. This properly accounts for the nesting of cells within each individual, and allows for valid statistical inference for the hypotheses of interest, such as assessing differential expression patterns with the `patternTest` or DE between the lineage endpoints with the `diffEndTest`, across all individuals in the dataset. More complex normalization procedures, that estimate variables to account for complex batch effects, such as RUV or ZINB-WaVE, can also be used as input to `tradeSeq`.

3.2 Minor ones

1. **The data visualization is weak throughout all the paper. For example, all the gene set enrichment results are in supplementary tables, which is not convenient and make paper difficult to read. Top or selected gene sets could be shown in main figures.**

The gene set enrichment results serve to demonstrate the relevance of the genes that are discovered by `tradeSeq`. We have avoided having to select gene sets from a long list of significant gene sets, and only show the top 20 gene sets in Supplementary Tables. We have therefore chosen to not show selected gene sets in main figures. Please also note that the gene set enrichment statistics in the olfactory epithelium case study are based on gene set overlap with the differentially expressed genes at a particular FDR level, and not on the ranked gene list, so we can show, e.g., the relevance of genes uniquely discovered by `tradeSeq` (Supplementary Table S3), or the top genes discovered by `tradeSeq` (Supplementary Table S1). Hence, this obviates the traditional gene set enrichment analysis plots.

2. **In general, the authors should focus on the `patternTest` function, which harbors the highest novelty in this paper, and provide sufficient evidences to demonstrate how robust it is.**

We believe that our manuscript provides valuable contributions beyond the `patternTest`. Indeed, `BEAM` and `GPfates` perform similar tests, though they are less powerful and limited to only two-lineage trajectories. The particular novelty of our manuscript is that we provide a range of interpretable hypothesis tests that are all built on the NB-GAM framework of `tradeSeq`. Moreover, `tradeSeq` is modular and hence may be used downstream of several trajectory inference methods, as shown in our simulation study. This opens the door to trajectory-based differential expression analysis downstream of future trajectory inference methodology that might perform even better than the current state-of-the-art. Furthermore, `tradeSeq` is capable of handling more than two lineages and uniquely allows to account for zero inflation.

References

- M. J. Nueda, S. Tarazona, and A. Conesa. Next maSigPro: updating maSigPro bioconductor package for RNA-seq time series. *Bioinformatics*, 30(18):2598–2602, 9 2014. ISSN 1367-4803. doi: 10.1093/bioinformatics/btu333. URL <https://academic.oup.com/bioinformatics/article-lookup/doi/10.1093/bioinformatics/btu333>.
- Tiziana Sanavia, Francesca Finotello, and Barbara Di Camillo. FunPat: function-based pattern analysis on RNA-seq time series data. *BMC Genomics*, 16(Suppl 6):S2, 2015. ISSN 1471-2164. doi: 10.1186/1471-2164-16-S6-S2. URL <http://www.ncbi.nlm.nih.gov/pubmed/26046293http://www.pubmedcentral.nih.gov/articlerender.fcgi?artid=PMC4460925http://bmcbgenomics.biomedcentral.com/articles/10.1186/1471-2164-16-S6-S2>.
- Rhonda Bacher, Ning Leng, Li-Fang Chu, Zijian Ni, James A. Thomson, Christina Kendzierski, and Ron Stewart. Trendy: segmented regression analysis of expression dynamics in high-throughput ordered profiling experiments. *BMC Bioinformatics*, 19(1):380, 12 2018. ISSN 1471-2105. doi: 10.1186/s12859-018-2405-x. URL <https://bmcbioinformatics.biomedcentral.com/articles/10.1186/s12859-018-2405-x>.

- Cole Trapnell, Davide Cacchiarelli, Jonna Grimsby, Prapti Pokharel, Shuqiang Li, Michael Morse, Niall J Lennon, Kenneth J Livak, Tarjei S Mikkelsen, and John L Rinn. The dynamics and regulators of cell fate decisions are revealed by pseudotemporal ordering of single cells. *Nature Biotechnology*, 32(4):381–386, 4 2014. ISSN 1087-0156. doi: 10.1038/nbt.2859. URL <http://www.ncbi.nlm.nih.gov/pubmed/24658644><http://www.pubmedcentral.nih.gov/articlerender.fcgi?artid=PMC4122333><http://www.nature.com/articles/nbt.2859>.
- Zhicheng Ji and Hongkai Ji. TSCAN: Pseudo-time reconstruction and evaluation in single-cell RNA-seq analysis. *Nucleic Acids Research*, 44(13):e117–e117, 7 2016. ISSN 0305-1048. doi: 10.1093/nar/gkw430. URL <https://academic.oup.com/nar/article-lookup/doi/10.1093/nar/gkw430>.
- Koen Van den Berge, Charlotte Soneson, Mark D. Robinson, and Lieven Clement. stageR: a general stage-wise method for controlling the gene-level false discovery rate in differential expression and differential transcript usage. *Genome Biology*, 18(1):151, 2017. ISSN 1474-760X. doi: 10.1186/s13059-017-1277-0. URL <http://www.ncbi.nlm.nih.gov/pubmed/28784146><http://www.pubmedcentral.nih.gov/articlerender.fcgi?artid=PMC5547545><http://genomebiology.biomedcentral.com/articles/10.1186/s13059-017-1277-0>.
- Wouter Saelens, Robrecht Cannoodt, Helena Todorov, and Yvan Saeys. A comparison of single-cell trajectory inference methods. *Nature Biotechnology*, page 1, 4 2019. ISSN 1087-0156. doi: 10.1038/s41587-019-0071-9. URL <http://www.nature.com/articles/s41587-019-0071-9>.
- Russell B. Fletcher, Diya Das, Levi Gadye, Kelly N. Street, Ariane Baudhuin, Allon Wagner, Michael B. Cole, Quetzal Flores, Yoon Gi Choi, Nir Yosef, Elizabeth Purdom, Sandrine Dudoit, Davide Risso, and John Ngai. Deconstructing Olfactory Stem Cell Trajectories at Single-Cell Resolution. *Cell Stem Cell*, 20(6):817–830, 6 2017. ISSN 19345909. doi: 10.1016/j.stem.2017.04.003. URL <http://www.ncbi.nlm.nih.gov/pubmed/28506465><http://www.pubmedcentral.nih.gov/articlerender.fcgi?artid=PMC5484588><https://linkinghub.elsevier.com/retrieve/pii/S1934590917301273>.
- Carolyn A. de Graaf, Jarny Choi, Tracey M. Baldwin, Jessica E. Bolden, Kirsten A. Fairfax, Aaron J. Robinson, Christine Biben, Clare Morgan, Kerry Ramsay, Ashley P. Ng, Maria Kauppi, Elizabeth A. Kruse, Tobias J. Sargeant, Nick Seidenman, Angela D’Amico, Marthe C. D’Ombra, Erin C. Lucas, Sandra Koernig, Adriana Baz Morelli, Michael J. Wilson, Steven K. Dower, Brenda Williams, Shen Y. Heazlewood, Yifang Hu, Susan K. Nilsson, Li Wu, Gordon K. Smyth, Warren S. Alexander, and Douglas J. Hilton. Haemopedia: An Expression Atlas of Murine Hematopoietic Cells. *Stem Cell Reports*, 7(3):571–582, 9 2016. ISSN 22136711. doi: 10.1016/j.stemcr.2016.07.007. URL <http://www.ncbi.nlm.nih.gov/pubmed/27499199><http://www.pubmedcentral.nih.gov/articlerender.fcgi?artid=PMC5031953><http://linkinghub.elsevier.com/retrieve/pii/S221367111630131X>.
- David S Fischer, Fabian J Theis, and Nir Yosef. Impulse model-based differential expression analysis of time course sequencing data. *Nucleic Acids Research*, 46(20):e119–e119, 8 2018. ISSN 0305-1048. doi: 10.1093/nar/gky675. URL <https://academic.oup.com/nar/advance-article/doi/10.1093/nar/gky675/5068248>.
- Michael I Love, Wolfgang Huber, and Simon Anders. Moderated estimation of fold change and dispersion for RNA-seq data with DESeq2. *Genome Biology*, 15(12):550, 12 2014. ISSN 1465-6906. doi: 10.1186/s13059-014-0550-8. URL <http://genomebiology.com/2014/15/12/550>.
- S. N. Wood. On p-values for smooth components of an extended generalized additive model. *Biometrika*, 100(1):221–228, 10 2012. ISSN 0006-3444. doi: 10.1093/biomet/ass048. URL <http://biomet.oxfordjournals.org/cgi/doi/10.1093/biomet/ass048>.
- Franziska Paul, Ya’ara Arkin, Amir Giladi, Diego Adhemar Jaitin, Ephraim Kenigsberg, Hadas Keren-Shaul, Deborah Winter, David Lara-Astiaso, Meital Gury, Assaf Weiner, Eyal David, Nadav Cohen, Felicia Kathrine Bratt Lauridsen, Simon Haas, Andreas Schlitzer, Alexander Mildner, Florent Ginhoux, Steffen Jung, Andreas Trumpp, Bo Torben Porse, Amos Tanay, and Ido Amit. Transcriptional Heterogeneity and Lineage Commitment in Myeloid Progenitors. *Cell*, 163(7):1663–1677, 12 2015. ISSN 0092-8674. doi: 10.1016/J.CELL.2015.11.013. URL <https://www.sciencedirect.com/science/article/pii/S0092867415014932?via%3Dihub#app3>.

- Simon N. Wood, Natalya Pya, and Benjamin Säfken. Smoothing Parameter and Model Selection for General Smooth Models. *Journal of the American Statistical Association*, 111(516):1548–1563, 10 2016. ISSN 0162-1459. doi: 10.1080/01621459.2016.1180986. URL <https://www.tandfonline.com/doi/full/10.1080/01621459.2016.1180986>.
- Vladimir Yu. Kiselev, Veronique Juvin, Mouhannad Malek, Nicholas Luscombe, Phillip Hawkins, Nicolas Le Novère, and Len Stephens. Perturbations of PIP3 signalling trigger a global remodelling of mRNA landscape and reveal a transcriptional feedback loop. *Nucleic Acids Research*, 43(20):gkv1015, 10 2015. ISSN 0305-1048. doi: 10.1093/nar/gkv1015. URL <https://academic.oup.com/nar/article-lookup/doi/10.1093/nar/gkv1015>.
- Davide Risso, Liam Purvis, Russell B. Fletcher, Diya Das, John Ngai, Sandrine Dudoit, and Elizabeth Purdom. clusterExperiment and RSEC: A Bioconductor package and framework for clustering of single-cell and other large gene expression datasets. *PLOS Computational Biology*, 14(9):e1006378, 9 2018a. ISSN 1553-7358. doi: 10.1371/journal.pcbi.1006378. URL <http://dx.plos.org/10.1371/journal.pcbi.1006378>.
- Xiaojie Qiu, Qi Mao, Ying Tang, Li Wang, Raghav Chawla, Hannah A Pliner, and Cole Trapnell. Reversed graph embedding resolves complex single-cell trajectories. *Nature Methods*, 8 2017. doi: 10.1038/nmeth.4402. URL <https://www.nature.com/nmeth/journal/vaop/ncurrent/full/nmeth.4402.html>.
- Andre Olsson, Meenakshi Venkatasubramanian, Viren K. Chaudhri, Bruce J. Aronow, Nathan Salomonis, Harinder Singh, and H. Leighton Grimes. Single-cell analysis of mixed-lineage states leading to a binary cell fate choice. *Nature*, 537(7622):698–702, 9 2016. ISSN 0028-0836. doi: 10.1038/nature19348. URL <http://www.ncbi.nlm.nih.gov/pubmed/27580035><http://www.pubmedcentral.nih.gov/articlerender.fcgi?artid=PMC5161694><http://www.nature.com/articles/nature19348>.
- Alexey Sergushichev. An algorithm for fast preranked gene set enrichment analysis using cumulative statistic calculation. *bioRxiv*, page 060012, 6 2016. doi: 10.1101/060012. URL <https://www.biorxiv.org/content/early/2016/06/20/060012>.
- Davis J McCarthy, Yunshun Chen, and Gordon K Smyth. Differential expression analysis of multifactor RNA-Seq experiments with respect to biological variation. *Nucleic acids research*, 40(10):4288–97, 5 2012. ISSN 1362-4962. doi: 10.1093/nar/gks042. URL <http://www.pubmedcentral.nih.gov/articlerender.fcgi?artid=3378882&tool=pmcentrez&rendertype=abstract>.
- Davide Risso, Liam Purvis, Russell Fletcher, Diya Das, John Ngai, Sandrine Dudoit, and Elizabeth Purdom. clusterExperiment and RSEC: A Bioconductor package and framework for clustering of single-cell and other large gene expression datasets. *bioRxiv*, page 280545, 3 2018b. doi: 10.1101/280545. URL <https://www.biorxiv.org/content/early/2018/03/12/280545>.

Reviewers' Comments:

Reviewer #1:

Remarks to the Author:

The authors have done a great job addressing my previous comments and I have no additional concerns.

Reviewer #2:

Remarks to the Author:

Review

In this revision, Van den Berge et al have included a large number of benchmarks and clarifications missing from the original submission.

The reason for excluding edgeR due to not performing shrinkage seems arbitrary, but the work is still substantial. If the authors would elect to also demonstrate to the reader the results of using edgeR it would further strengthen the manuscript, but not required in the current state.

Similarly, it would have been better if the clustering method had been compared to k-medoids on the fitted tradeSeq parameters (using the same R package as Monocle wraps). It is of interest to a reader to see how RSEC on the principal components compares of fitted parameters compares to RSEC on the parameters directly, as well as k-medoids on the parameters directly. However, since the pattern clustering is not the primary aim of the manuscript, this can also be considered optional.

Beside these points, the other issues have been addressed.

Reviewer #3:

Remarks to the Author:

The authors provide reasonable changes and explanation for the major and minor issues. The revision is much improved. However, I still have one major concern about real data application. Although the proposed statistical method is very valuable for association analysis after trajectory analysis, the significance and application is limited by the two case studies, which are from mouse studies using old single cell techniques (MARS-Seq and SMART-seq). Given the fact that Droplet-based single cell technique, particularly 10X Genomics Chromium system, has become a major platform in the single cell field, it will be much more useful and attractive if the author can show how it works on 10X data from human. In fact, there are many public 10X datasets (e.g, Wouter Saelens et al. Nature Biotechnology, 2019 and <https://support.10xgenomics.com/single-cell-gene-expression/datasets>). In addition, the zero-inflation problem is more severe in 10X data and the proposed method may have more advantages in this scenario.

Rebuttal: Trajectory-based differential expression analysis for single-cell sequencing data

November 28, 2019

1 Reviewer 1

The authors have done a great job addressing my previous comments and I have no additional concerns.

2 Reviewer 2

In this revision, Van den Berge et al have included a large number of benchmarks and clarifications missing from the original submission.

1. The reason for excluding edgeR due to not performing shrinkage seems arbitrary, but the work is still substantial. If the authors would elect to also demonstrate to the reader the results of using edgeR it would further strengthen the manuscript, but not required in the current state.

We feel that incorporating a full comparison against edgeR with a basis function design matrix would distract the reader from the main message of our manuscript, which is the novel diversity of statistical testing procedures to interpret dynamic datasets, implemented in a general and modular package that avoids ‘pipeline lock-in’. While we wish to compare our method against top performing methods from the literature, we think it is outside the scope of this manuscript to develop methods that are not described in the literature and compare against them.

We do emphasize that using edgeR, instead of mgcv, could be an alternative approach to fitting the NB-GAMs that we propose in our manuscript. We note, however, that the tests implemented in tradeSeq are implemented based on the tradeSeq or mgcv models output. If edgeR were used to fit the smoothers, one would still be limited by the inference procedures developed within the edgeR framework for (discrete) differential expression testing. While it would be possible to re-implement the DE tests implemented in tradeSeq to work with edgeR output, we consider this to be out of the scope of this manuscript.

However, we note that the associationTest implemented by tradeSeq can be easily implemented in an edgeR framework by testing whether all basis function parameters of a smoother are equal to zero, and we have now included this evaluation in the simulation study on the cyclic topology. There, the performance is similar to a tradeSeq-based associationTest, confirming that edgeR would be just another vehicle to estimate the NB-GAMs proposed in our manuscript. We have now added this to the Methods Section where we describe the edgeR method, on lines 335-347:

“ edgeR [McCarthy et al., 2012] is a discrete differential expression method, where the groups under comparison must be defined a priori. It is therefore useful for assessing DE between, for example, annotated clusters or different treatment groups. For such comparisons, edgeR is a powerful method with high sensitivity. Note that, while edgeR was originally developed for group-based differential expression, it would be possible to incorporate the basis functions of the smoothers as continuous covariates in the

model. However, no regularization would be performed on the estimation of the smoother regression coefficients, and the model would be prone to overfitting. A similar approach was evaluated in Fischer et al. [2018], where `DESeq2` [Love et al., 2014] was used to fit splines by incorporating natural cubic basis functions in the linear predictor. Hence, while it is possible to fit smoothers by using `edgeR`, instead of `mgcv`, we emphasize that this would merely be an alternative approach to fitting the NB-GAMs we propose in our manuscript. Indeed, `edgeR` does not provide an implementation of the DE tests in `tradeSeq`. Only the `associationTest` is readily available in `edgeR` by testing whether all basis function parameters are equal to zero and the other tests would require a similar development as presented in `tradeSeq`. ”

We have updated the Results for the simulation study on datasets with a cyclic topology to include the `edgeR`-based `associationTest` on lines 482-486:

“Finally, we evaluate an `edgeR`-based `associationTest` through fitting the NB-GAMs with `edgeR` instead of with `mgcv` (method `edgeR_assoc`, see Methods for details), and note that its performance is similar to the `tradeSeq associationTest` (Supplementary Figure S7). This could be expected because few basis functions were selected for this simulation setting. In applications that require a rich basis, however, the `edgeR` implementation will be prone to overfitting.”

Besides the practical arguments mentioned above, there are also more technical arguments for not fitting the smoothers with `edgeR`. As mentioned before, since there is no penalization implemented in `edgeR`, the smoothers would likely be prone to overfitting, especially if many basis functions are fitted, which is not the case in the simulation studies. Related, if we are indeed fitting many basis functions, `edgeR` cannot deal with the aliasing (i.e., linear dependencies) that may exist between them, which renders inverting the design matrix, e.g., for calculating the variance on the basis function parameter estimates $\hat{\beta}_{g1k}$, problematic. Indeed, fitting the NB-GAMs using `edgeR` on the multifurcating trajectory simulated dataset errors because of aliasing issues as soon as more than 7 knots are used, while the `mgcv` package is able to deal with this.

Supplementary Figure S7: *Simulation study results, including edgeR-based associationTest for the cyclic scenario.* PCA plots for the (a) cyclic, (b) bifurcating, and (c) multifurcating simulated trajectories. The plotting symbol for each cell is colored according to its true pseudotime; trajectories (in black) were inferred by printrcurve in (a) and slingshot in (b) and (c). (d-f) Scatterplot of the true positive rate (TPR) vs. the false discovery rate (FDR) or false discovery proportion (FDP) for various DE methods applied to the simulated datasets. Panel (d) displays the average performance curves of DE methods across seven out of 10 cyclic datasets that worked on all DE methods (Monocle 3 errored on three datasets). The associationTest from tradeSeq has superior performance for discovering genes whose expression is associated with pseudotime, as compared to Monocle 3. When investigating differential expression between lineages of a trajectory, the patternTest of tradeSeq consistently outperforms the diffEndTest across all three TI methods, since it is capable of comparing expression across entire lineages. Panel (e) displays the average performance curves across the three bifurcating datasets where all TI methods recovered the correct topology. Here, all tradeSeq patternTest workflows, tradeSeq_slingshot_end, and have similar performance and all are superior to BEAM, ImpulseDE2, and GPfates. Note that the performance of tradeSeq_Monocle2_end deteriorates as compared to tradeSeq_slingshot_end; the curve for tradeSeq_GPfates_end is not visible in this panel due to its low performance. For the multifurcating dataset of panel (f), tradeSeq_slingshot has the highest performance, closely followed by tradeSeq_Monocle2 and edgeR. Source data are provided as a Source Data file.

2. Similarly, it would have been better if the clustering method had been compared to k-medoids on the fitted tradeSeq parameters (using the same R package as Monocle wraps). It is of interest to a reader to see how RSEC on the principal components compares of fitted parameters compares to RSEC on the parameters directly, as well as k-medoids on the parameters directly. However, since the pattern clustering is not the primary aim of the manuscript, this can also be considered optional.

We agree that it is of interest to benchmark different procedures for clustering genes, also for other

applications, e.g., gene co-expression analyses. We have therefore evaluated the stability of gene clusters on the dataset from Paul et al. [2015]. As in the manuscript, we cluster the `tradeSeq` fitted values, and we benchmark three different approaches: RSEC based on PCA of the fitted values (default RSEC implementation), RSEC based on the fitted values directly, and PAM clustering as implemented in the `cluster` package, also used by `Monocle`.

While comparing clusterings with different numbers of preset clusters using criteria such as the average silhouette width is appropriate, this is inappropriate for comparing different clustering methods, as it is biased in favor of clustering procedures that use the same (dis)similarity measure as the silhouette width. We therefore resort to non-parametric bootstrapping of the cells and assess the stability of clusterings over the bootstrap samples, as compared to the original clustering based on the full dataset, using the adjusted Rand index (ARI) [Hubert and Arabie, 1985]. We have now added a sentence to the ‘Gene expression families.’ paragraph in the Results section, on lines 653-655, and a corresponding Supplementary Figure, which we also show below.

“ In general, we found that RSEC clustering provides a more stable clustering than Partitioning around medoids (PAM) (Supplementary Figure S20), the latter of which is also used by Monocle to cluster genes.”

Supplementary Figure S20: *Stability of clustering methods across bootstraps*. Boxplots (center line, median; box limits, upper and lower quartiles; whiskers, $1.5 \times$ interquartile range) of the clustering stability are shown. Clustering stability is evaluated using non-parametric bootstrapping based on all genes found to be significant on a 5% FDR level by the `patternTest` in the case study dataset from Paul et al. [2015]. For each iteration, cells are sampled with replacement, and we refit the NB-GAM using `tradeSeq`, after which we cluster genes based on the `tradeSeq` fitted values, using Partitioning around medoids (PAM), and RSEC. We compare against PAM since this method is also used in Monocle for gene clustering. For RSEC clustering, we evaluate both clustering on the fitted values directly (method ‘RSEC_noDR’ in the Figure) as well as clustering after dimensionality reduction (method ‘RSEC’ in the Figure) with principal component analysis (the default for the RSEC method as implemented in `clusterExperiment`, the number of retained principal components are automatically determined by the method). The stability of the clustering is evaluated by comparing the bootstrapped clusterings with the original clustering based on the full dataset using the Adjusted Rand Index (ARI) [Hubert and Arabie, 1985]. For computational reasons, we restricted this evaluation to six bootstrap iterations.

Beside these points, the other issues have been addressed.

3 Reviewer 3

The authors provide reasonable changes and explanation for the major and minus issues. The revision is much improved. However, I still have one major concern about real data application. Although the proposed statistical method is very valuable for association analysis after trajectory analysis, the significance and application is limited by the two case studies, which are from mouse studies using old single cell techniques (MARS-Seq and SMART-seq). Given the fact that Droplet-based single cell technique, particularly 10X Genomics Chromium

system, has become a major platform in the single cell field, it will be much more useful and attractive if the author can show how it works on 10X data from human. In fact, there are many public 10X datasets (e.g, Wouter Saelens et al. Nature Biotechnology, 2019 and <https://support.10xgenomics.com/single-cell-gene-expression/datasets>). In addition, the zero-inflation problem is more severe in 10X data and the proposed method may have more advantages in this scenario.

We have now incorporated a fourth case study dataset on adipocyte differentiation from the 10x Genomics platform. The data were recently published in Merrick et al. [2019]. We have fitted the trajectory using *slingshot* and performed differential expression analysis using *tradeSeq*, leading to interesting discoveries in gene expression patterns along the lineages of the trajectory. In the analysis, we do not account for zero inflation, since it has previously been shown that accounting for zero inflation is not beneficial for 10x datasets [Van den Berge et al., 2018, Svensson, 2019]. Indeed, although there are many zeros in 10x datasets, the positive counts are also low, which hampers zero inflation detection since both zeros and low positive counts typically fit well under a (non zero-inflated) negative binomial distribution. In the revised manuscript, we have now updated the text in the Methods and Results Sections to include the 10x adipocyte differentiation dataset, on lines 415-427 (Methods) and 745-760 (Results), which we also show below, including the corresponding Figures.

Methods:

*“As final case study, we analyze a 10X adipocyte differentiation dataset described in Merrick et al. [2019]. The gene expression counts were downloaded from GEO with accession number GSE128889. We focus the analysis on the single cells collected from 12-day old mice. The raw dataset consists of 27,998 genes and 11,423 cells. We only retain genes with a count of at least 2 in at least 400 cells, and normalize the data using full quantile normalization [Bolstad et al., 2003]. Since not all cells in the dataset are involved in the adipocyte differentiation process, we first identify the relevant clusters of cells using the marker genes described in the original manuscript. We adopt k-means clustering ($k = 10$) on the top 8 principal components of log-transformed counts. Using this clustering, we identify the relevant clusters based on the reported markers, and subsequently adopt UMAP dimensionality reduction [Becht et al., 2019, McInnes et al., 2018] on the top 20 principal components for that subset of cells. The processed dataset consists of 2,851 genes and 8,071 cells. We use *slingshot* [Street et al., 2018] for trajectory inference in 2D UMAP space.”*

Results:

*“As final case study, we reanalyze a 10X scRNA-seq dataset from Merrick et al. [2019], studying adipocyte differentiation from the developing sub-cutaneous inguinal white adipose tissue (iWAT) of 12-day-old mice. We use *tradeSeq* to fit NB-GAMs with 8 knots (Supplementary Figure S26) based on the trajectory inferred by *slingshot* in 2D UMAP space. Similar to the original manuscript, the progenitor cells differentiate in two different cell populations (Supplementary Figure S27). While we confirm *Dpp4+* and *Wnt2* as interstitial progenitor markers, we discover several other markers as top genes of our *startVsEndTest* procedure that are even more pronounced, e.g. *Pi16*, *Akr1c18*, *Fn1* and *Fbn1* (Supplementary Figure S28). In addition, we search for markers differentiating the two differentiated cell populations. Since these are relatively large, heterogeneous groups of cells, the *diffEndTest* is not representative for the entire set of cells, however, the *earlyDETest* can be used to discover DE across their developmental range. This reveals several interesting patterns, such as genes upregulated in the adipocyte precursor stage, and subsequently downregulated in only a single differentiated cell population (e.g. *Mgp* and *Meox2*; Supplementary Figure S29), as well as genes that are sporadically highly expressed across the entire lineage for one of the two differentiated cell populations (e.g. *H19* and *Col14a1*; Supplementary Figure S30).”*

Supplementary Figure S26: 10X case study: *Selecting the optimal number of knots k using the AIC.* Selecting the optimal number of knots, $k \in \{3, \dots, 10\}$, using the AIC for a random subset of 200 genes, as implemented in the `evaluateK` function in `tradeSeq`. The left panel shows boxplots (center line, median; box limits, upper and lower quartiles; whiskers, $1.5 \times$ interquartile range) of the differences in AIC value with respect to the gene-wise average AIC for the range of k . The middle panels show the evolution of the average AIC (second panel) and relative AIC (third panel) across k . The relative AIC is defined as the relative change with respect to the average AIC at $k = 3$. The barplot in the right panel shows the number of genes which achieve their lowest AIC value for a given k . Here, only genes for which the AIC value varied substantially enough across k (i.e., range in AIC greater than 2) are considered.

Supplementary Figure S27: *10X case study: Inferred trajectory*. The scRNA-seq data is plotted in 2D UMAP space, and each cell is colored according to its cluster membership as derived by k-means clustering with $k = 6$ clusters. The black solid line represents the trajectory as estimated by *slingshot*.

Supplementary Figure S28: 10X case study: Top markers for progenitor cell population. The scRNA-seq data is plotted in 2D UMAP space, and each cell is colored according to the expression of the corresponding gene, divided in 4 bins, where blue corresponds to low expression and red corresponds to high expression. The top row corresponds to two marker genes, *Dpp4+* and *Wnt2* from the original manuscript [Merrick et al., 2019]. Other plots are top genes identified with the `startVsEndTest` procedure in `tradeSeq`.

Supplementary Figure S29: 10X case study: *Genes upregulated in the adipocyte precursor stage and a single differentiated cell type.* The scRNA-seq data is plotted in 2D UMAP space, and each cell is colored according to the expression of the corresponding gene, divided in 4 bins, where blue corresponds to low expression and red corresponds to high expression.

Supplementary Figure S30: 10X case study: Genes sporadically upregulated across the entire lineage for a single differentiated cell type. The scRNA-seq data is plotted in 2D UMAP space, and each cell is colored according to the expression of the corresponding gene, divided in 4 bins, where blue corresponds to low expression and red corresponds to high expression.

References

- Davis J McCarthy, Yunshun Chen, and Gordon K Smyth. Differential expression analysis of multifactor RNA-Seq experiments with respect to biological variation. *Nucleic acids research*, 40(10):4288–97, 5 2012. ISSN 1362-4962. doi: 10.1093/nar/gks042. URL <http://www.pubmedcentral.nih.gov/articlerender.fcgi?artid=3378882&tool=pmcentrez&rendertype=abstract>.
- David S Fischer, Fabian J Theis, and Nir Yosef. Impulse model-based differential expression analysis of time course sequencing data. *Nucleic Acids Research*, 46(20):e119–e119, 8 2018. ISSN 0305-1048. doi: 10.1093/nar/gky675. URL <https://academic.oup.com/nar/advance-article/doi/10.1093/nar/gky675/5068248>.
- Michael I Love, Wolfgang Huber, and Simon Anders. Moderated estimation of fold change and dispersion for RNA-seq data with DESeq2. *Genome Biology*, 15(12):550, 12 2014. ISSN 1465-6906. doi: 10.1186/s13059-014-0550-8. URL <http://genomebiology.com/2014/15/12/550>.
- Franziska Paul, Ya’ara Arkin, Amir Giladi, Diego Adhemar Jaitin, Ephraim Kenigsberg, Hadas Keren-Shaul, Deborah Winter, David Lara-Astiaso, Meital Gury, Assaf Weiner, Eyal David, Nadav Cohen, Felicia Kathrine Bratt Lauridsen, Simon Haas, Andreas Schlitzer, Alexander Mildner, Florent Ginhoux, Steffen Jung, Andreas Trumpp, Bo Torben Porse, Amos Tanay, and Ido Amit. Transcriptional Heterogeneity and Lineage Commitment in Myeloid Progenitors. *Cell*, 163(7):1663–1677, 12 2015. ISSN 0092-8674. doi: 10.1016/J.CELL.2015.11.013. URL <https://www.sciencedirect.com/science/article/pii/S0092867415014932?via%3Dihub#app3>.
- Lawrence Hubert and Phipps Arabie. Comparing partitions. *Journal of Classification*, 2(1):193–218, 12 1985. ISSN 0176-4268. doi: 10.1007/BF01908075. URL <http://link.springer.com/10.1007/BF01908075>.

- David Merrick, Alexander Sakers, Zhazira Irgebay, Chihiro Okada, Catherine Calvert, Michael P Morley, Ivona Percec, and Patrick Seale. Identification of a mesenchymal progenitor cell hierarchy in adipose tissue. *Science (New York, N.Y.)*, 364(6438), 4 2019. ISSN 1095-9203. doi: 10.1126/science.aav2501. URL <http://www.ncbi.nlm.nih.gov/pubmed/31023895><http://www.pubmedcentral.nih.gov/articlerender.fcgi?artid=PMC6816238>.
- Koen Van den Berge, Fanny Perraudeau, Charlotte Soneson, Michael I. Love, Davide Risso, Jean-Philippe Vert, Mark D. Robinson, Sandrine Dudoit, and Lieven Clement. Observation weights unlock bulk RNA-seq tools for zero inflation and single-cell applications. *Genome Biology*, 19(1):24, 12 2018. ISSN 1474-760X. doi: 10.1186/s13059-018-1406-4. URL <https://genomebiology.biomedcentral.com/articles/10.1186/s13059-018-1406-4>.
- Valentine Svensson. Droplet scRNA-seq is not zero-inflated. *bioRxiv*, page 582064, 3 2019. doi: 10.1101/582064. URL <https://www.biorxiv.org/content/10.1101/582064v1>.
- B M Bolstad, R A Irizarry, M Astrand, and T P Speed. A comparison of normalization methods for high density oligonucleotide array data based on variance and bias. *Bioinformatics (Oxford, England)*, 19(2): 185–93, 1 2003. ISSN 1367-4803. URL <http://www.ncbi.nlm.nih.gov/pubmed/12538238>.
- Etienne Becht, Leland McInnes, John Healy, Charles-Antoine Dutertre, Immanuel W H Kwok, Lai Guan Ng, Florent Gehroux, and Evan W Newell. Dimensionality reduction for visualizing single-cell data using UMAP. *Nature Biotechnology*, 37:38–44, 12 2019. ISSN 1087-0156. doi: 10.1038/nbt.4314. URL <http://www.nature.com/doi/10.1038/nbt.4314>.
- Leland McInnes, John Healy, and James Melville. UMAP: Uniform Manifold Approximation and Projection for Dimension Reduction. *ArXiv*, 2 2018. URL <http://arxiv.org/abs/1802.03426>.
- Kelly Street, Davide Risso, Russell B. Fletcher, Diya Das, John Ngai, Nir Yosef, Elizabeth Purdom, and Sandrine Dudoit. Slingshot: cell lineage and pseudotime inference for single-cell transcriptomics. *BMC Genomics*, 19(1):477, 12 2018. ISSN 1471-2164. doi: 10.1186/s12864-018-4772-0. URL <https://bmcgenomics.biomedcentral.com/articles/10.1186/s12864-018-4772-0>.

Reviewers' Comments:

Reviewer #2:

Remarks to the Author:

The remaining issues have been resolved by the authors.

Reviewer #3:

Remarks to the Author:

Authors have addressed my comments by adding a new case study on 10X data. The results are expected and consistent with other datasets. I have no further comments.

---

## [Peer Review File · Nature Communications]

Reporting Summary

Nature Research wishes to improve the reproducibility of the work that we publish. This form provides structure for consistency and transparency in reporting. For further information on Nature Research policies, see Authors & Referees and the Editorial Policy Checklist.

Statistics

For all statistical analyses, confirm that the following items are present in the figure legend, table legend, main text, or Methods section.

n/a Confirmed

- | | | |
|-------------------------------------|-------------------------------------|--|
| [ ] | [x] | The exact sample size (n) for each experimental group/condition, given as a discrete number and unit of measurement |
| [x] | [ ] | A statement on whether measurements were taken from distinct samples or whether the same sample was measured repeatedly |
| [ ] | [x] | The statistical test(s) used AND whether they are one- or two-sided
Only common tests should be described solely by name; describe more complex techniques in the Methods section. |
| [x] | [ ] | A description of all covariates tested |
| [ ] | [x] | A description of any assumptions or corrections, such as tests of normality and adjustment for multiple comparisons |
| [ ] | [x] | A full description of the statistical parameters including central tendency (e.g. means) or other basic estimates (e.g. regression coefficient) AND variation (e.g. standard deviation) or associated estimates of uncertainty (e.g. confidence intervals) |
| [x] | [ ] | For null hypothesis testing, the test statistic (e.g. F , t , r) with confidence intervals, effect sizes, degrees of freedom and P value noted
Give P values as exact values whenever suitable. |
| [x] | [ ] | For Bayesian analysis, information on the choice of priors and Markov chain Monte Carlo settings |
| [x] | [ ] | For hierarchical and complex designs, identification of the appropriate level for tests and full reporting of outcomes |
| [ ] | [x] | Estimates of effect sizes (e.g. Cohen's d , Pearson's r), indicating how they were calculated |

Our web collection on statistics for biologists contains articles on many of the points above.

Software and code

Policy information about availability of computer code

Data collection

The simulated data was generated with the dyno R package, version 0.1.0 and the dynverse package, version 0.1. The mouse bone marrow case study dataset was downloaded from http://trapnell-lab.gs.washington.edu/public_share/valid_subset_GSE72857_cds2.RDS. The data is now available in our tradeSeqPaper GitHub repository, at <https://github.com/statOmics/tradeSeqPaper>. The olfactory epithelium dataset was downloaded from GEO with accession number GSE95601.

Data analysis

Data has been analyzed using R packages tradeSeq v0.9.0, edgeR v3.22.5, Monocle 3 v2.99.3, zinbwave v 1.2.0, Monocle 2 v2.9.0, slingshot v1.1.3, mgcv v1.8-23, stageR v1.5.1, clusterExperiment v2.2.0, ggplot2 v3.1.0, tidyverse v1.2.1, fgsea 1.6.0 and Python v3.6.7 module GPfates v1.0.0.

For manuscripts utilizing custom algorithms or software that are central to the research but not yet described in published literature, software must be made available to editors/reviewers. We strongly encourage code deposition in a community repository (e.g. GitHub). See the Nature Research guidelines for submitting code & software for further information.

Data

Policy information about availability of data

All manuscripts must include a data availability statement. This statement should provide the following information, where applicable:

- Accession codes, unique identifiers, or web links for publicly available datasets
- A list of figures that have associated raw data
- A description of any restrictions on data availability

The code to generate all simulated datasets is included in the GitHub repository of the paper at <https://github.com/statOmics/tradeSeqPaper>. The data for the bulk RNA-seq time course is available at https://github.com/376-daniel-spies/rna-seq_tcComp. The data for the mouse bone marrow case study were downloaded from http://trapnell-lab.gs.washington.edu/public_share/valid_subset_GSE72857_cds2.RDS and is now available in our GitHub repository at <https://github.com/statOmics/tradeSeqPaper>. The raw data for the olfactory epithelium case study are available on GEO with accession number GSE95601. The data for the 10X adipocyte case study are available on GEO with accession number GSE128889.

Field-specific reporting

Please select the one below that is the best fit for your research. If you are not sure, read the appropriate sections before making your selection.

Life sciences Behavioural & social sciences Ecological, evolutionary & environmental sciences

For a reference copy of the document with all sections, see [nature.com/documents/nr-reporting-summary-flat.pdf](https://www.nature.com/documents/nr-reporting-summary-flat.pdf)

Life sciences study design

All studies must disclose on these points even when the disclosure is negative.

Sample size	This study has re-used public data, and no sample size calculations were performed.
Data exclusions	In the mouse bone marrow case study, the dendritic and eosinophil cell types were removed prior to the analysis, since these did not seem to contribute to the trajectory. In the mouse olfactory epithelium case study, only genes with at least five counts per million in at least 15 cells were retained for analysis. In the adipocyte differentiation case study, we only retain cells relevant to the trajectory using marker genes defined in the original manuscript, we only retain genes with a count of at least 2 in at least 400 cells.
Replication	Our study replicates findings of previous studies on public datasets.
Randomization	Not applicable for this study, since only public datasets were used.
Blinding	Not applicable for this study, since only public datasets were used.

Reporting for specific materials, systems and methods

We require information from authors about some types of materials, experimental systems and methods used in many studies. Here, indicate whether each material, system or method listed is relevant to your study. If you are not sure if a list item applies to your research, read the appropriate section before selecting a response.

Materials & experimental systems

n/a	Involvement in the study
[x]	[ ] Antibodies
[x]	[ ] Eukaryotic cell lines
[x]	[ ] Palaeontology
[x]	[ ] Animals and other organisms
[x]	[ ] Human research participants
[x]	[ ] Clinical data

Methods

n/a	Involvement in the study
[x]	[ ] ChIP-seq
[x]	[ ] Flow cytometry
[x]	[ ] MRI-based neuroimaging